# Tucker Attention: A generalization of approximate attention mechanisms

Timon Klein [* 1]   Jonas Kusch [* 2]   Sebastian Sager [1 3]   Stefan Schnake [* 4]   Steffen Schotthöfer [* 4]

## Abstract

The pursuit of reducing the memory footprint of the self-attention mechanism in multi-headed self attention (MHA) spawned a rich portfolio of methods, e.g., group-query attention (GQA) and multi-head latent attention (MLA). The methods leverage specialized low-rank factorizations across embedding dimensions or attention heads. From the point of view of classical low-rank approximation, these methods are unconventional and raise questions of which objects they really approximate and how to interpret the low-rank behavior of the resulting representations. To answer these questions, this work proposes a generalized view on the weight objects in the self-attention layer and a factorization strategy, which allows us to construct a parameter efficient scheme, called Tucker Attention. Tucker Attention requires an order of magnitude fewer parameters for comparable validation metrics, compared to GQA and MLA, as evaluated in LLM and ViT test cases. Additionally, Tucker Attention encompasses GQA, MLA, MHA as special cases and is fully compatible with flash-attention and rotary position embeddings (RoPE). This generalization strategy yields insights of the actual ranks achieved by MHA, GQA, and MLA, and further enables simplifications for MLA. Source code available at https://github.com/ScSteffen/Tucker-Attention.

## 1. Introduction

The development of the attention mechanism (Bahdanau et al., 2014) and its use in transformer architectures (Vaswani et al., 2017) has significantly impacted deep learning. Transformer architectures, relying on multi-head attention (MHA) and related methods, however, incur substantial computational costs. MHA introduces a large number of parameters in the key, query, value, and output weight matrices, which induces substantial memory costs during pre-training when storing optimizer states and gradients.

These memory costs manifest at multiple stages: during training through optimizer states and gradients, during inference through the KV-cache (whose size scales with the number of heads and head dimension), and during attention computation through GPU cache pressure in kernels like flash-attention (Dao et al., 2022). Designing compressed, parameter-efficient representations of attention weights that address all three bottlenecks simultaneously remains an open challenge.

A main approach to reduce KV cache overhead and the overall parameter count in transformer architectures is to leverage parameter redundancies and the limited utility of individual attention heads. A large amount of work has studied such redundancies, which are summarized below. (Voita et al., 2019) shows that many heads in multi-head attention contribute little and can be pruned, implying the existence of low-rank structure across heads. (Brunner et al., 2020) demonstrates that portions of the attention weights may not affect the model output. (Bhojanapalli et al., 2020) identifies head dimensions that induce a low-rank bottleneck preventing the representation of arbitrary context vectors. (Behnke & Heafield, 2020) proposes head pruning during training. (Li et al., 2018) mitigates redundant learning across heads through disagreement regularization. (Li et al., 2021) applies differentiable subset pruning to remove redundant head information. (Michel et al., 2019) observes that some layers contain many redundant heads, whereas others rely heavily on a large head dimension.

Several methods have been proposed to leverage parameter redundancies. Among the most widely used approaches is Multi-Query Attention (MQA), which employs a single head for the key and value projections (Shazeer, 2019). MQA lowers the parameter count and enables efficient KV

---

[*]Equal contribution, shared first and last authorship. [1]Department of Mathematics; Otto von Guericke University Magdeburg; 39106 Magdeburg; Germany. [2]Scientific Computing, Norwegian University of Life Sciences, Ås, Norway [3]Max Planck Institute for Dynamics of Complex Technical Systems; 39106 Magdeburg; Germany. [4]Computer Science and Mathematics Division, Oak Ridge National Laboratory, Oak Ridge, TN 37980, USA. Correspondence to: Steffen Schotthöfer <schotthofers@ornl.gov>.

*Proceedings of the 43$^{rd}$ International Conference on Machine Learning*, Seoul, South Korea. PMLR 306, 2026. Copyright 2026 by the author(s).

caching, where the cache size is independent of the number of attention heads. To trade off parameter efficiency and accuracy, Grouped-Query Attention (GQA) (Ainslie et al., 2023) adopts a limited number of heads for the key and value projections, achieving improved accuracy over MQA. A further direction is Multi-Head Latent Attention (MLA) (Liu et al., 2024), which compresses the query, key, and value weights in low-rank format, resulting in reduced parameter counts. Moreover, the KV-cache of MLA that scales only with the low-rank latent dimension, instead of the per-head embedding dimension and number of heads.

In recent years, several works have investigated the low-rank structure of attention weights through approaches similar to low-rank adapters (LoRA) for fine-tuning (Hu et al., 2022). (Cordonnier et al., 2020) leverages low-rank structure in multi-head attention to enforce information sharing by learning a mixing vector that selects relevant information from shared projection maps. Related to the MLA approach, in (Zhang et al., 2025) query, key, and value weights are individually represented through a low-rank canonical polyadic (CP) decomposition while maintaining compatibility with position encodings like rotational position embedding (RoPE) (Su et al., 2023). Similarly, (Gu et al., 2025) represents query, key, value, and output weights as an order three tensor, and then stacks these as an order four tensor before factorizing the object in low-rank Tucker format. (Meng et al., 2025) successfully represented GQA layers as MLA layers, which motivates further generalization.

We remark that there is a wealth of approximate attention methods that avoid the quadratic cost in sequence length induced by MHA and similar methods, e.g., linear attention (Wang et al., 2020). These methods invoke a different efficiency to accuracy trade-off discussion that is out of scope for this paper. This includes sparse attention mechanisms (Beltagy et al., 2020; Zaheer et al., 2020) that focus on subselecting chunks of the input sequence to enforce sparsity of the pre-softmax attention matrix. These methods are orthogonal to the proposed work and can be combined.

### 1.1. Contribution

In this paper, we provide a principled, interpretable, and generalized view on the attention weight representation. Instead of inspecting and manipulating the query, key, value and output matrices *individually*, we analyze the pre-softmax attention weights (query and key) and the post-softmax attention weights (value and output) as standalone objects. The former constitutes a bilinear form on the token sequence that yields the pre-softmax attention matrix per head, and the latter a linear form to combine the post-softmax attention matrices to the attention output, relative to the current input sequence. Thus, pre- and post-softmax attention weights

are each structured in an order three tensor object each.

As a corollary we observe that MHA itself is a special factorization of this object. We ask the question:

- Which tensor ranks does MHA assume in this context?

- Instead of approximating MHA, as the listed related works do, is it more efficient to approximate the tensor structure directly?

We find that MHA is the most efficient factorization if we assume pre- and post- softmax tensors to be of full rank. Under the reasonable assumption (Voita et al., 2019; Brunner et al., 2020) that there is low-rank structure in the pre- and post-softmax tensors, we propose a *direct* low-rank Tucker decomposition of those objects, simply called Tucker Attention.

This proposed Tucker Attention has two key features:

- **Expressive**: MHA, MQA, GQA, and MLA are all special cases of Tucker Attention corresponding to specific Tucker ranks. This unifying view reveals that none of these methods compress the head mode of the attention tensor, leaving exploitable redundancies (Section 2.2).

- **Informative**: Unlike prior methods that target individual weight matrices, Tucker Attention enables independent rank selection along head, query, key, value, and output modes. This flexibility yields up to $9\times$ parameter reduction at comparable accuracy (Section 4).

In Section 3, we show that the proposed Tucker Attention is compatible with KV caching, a variation of RoPE (Su et al., 2023), and Flash-Attention. As a corollary, we show that MLA is compatible RoPE under the proposed variation, even without the auxiliary construct of decoupled RoPE, as proposed by (DeepSeek-AI et al., 2024). Thus MLA with RoPE can be simplified significantly. Furthermore, compatibility with existing flash-attention implementations unifies the advantages of MQA and MLA to reduce memory pressure for flash-attention. Tucker Attention loads one key and value embedding vector for all heads of the query embeddings, just like MQA. By controlling the rank of the key and value embedding, Tucker Attention allows direct control of the KV-cache, and thus the ability to control the GPU cache pressure like MLA in inference mode. We remark that Tucker Attention enjoys these gains during training and inference.

Finally, in Section 4, we present results on a vision transformer transfer learning scenario and GPT2 pretraining. The results show Tucker Attention yields a factorization that is often by an order of magnitude more parameter efficient than GQA, MLA, or related methods[1].

---

[1]We define parameter-efficiency in this context as validation

**Notation** We use the following notation: vectors and scalars lowercase in italic font $(x)$, matrices in uppercase italic font $(W)$, and tensors in uppercase calligraphic font $(\mathcal{C})$. Matrix unfolding, a.k.a. matricization, of a tensor in mode $j$ is denoted by $\mathrm{Mat}_j(\cdot)$, and the n-mode product in mode $j$ is denote by $\times_j$. Both of these operations are defined in (Kolda & Bader, 2009)[Section 2].

## 2. Tensor attention reformulation

Our strategy is to rewrite the multi-head attention formalism as tensor-valued operations, which allows us to define flexible low-rank training methods for various types of attention formalisms. Let $X \in \mathbb{R}^{N \times d_{\mathrm{model}}}$ be an embedded sequence of $N$ tokens with embedding dimension $d_{\mathrm{model}}$. Standard MHA splits the $d_{\mathrm{model}} \times d_{\mathrm{model}}$ key, query, and value matrices across $n_{\mathrm{H}}$ heads by a column splitting:

$$W^{\mathrm{Q}} = [W_1^{\mathrm{Q}}, \cdots, W_{n_{\mathrm{H}}}^{\mathrm{Q}}],$$
$$W^{\mathrm{K}} = [W_1^{\mathrm{K}}, \cdots, W_{n_{\mathrm{H}}}^{\mathrm{K}}],$$
$$W^{\mathrm{V}} = [W_1^{\mathrm{V}}, \cdots, W_{n_{\mathrm{H}}}^{\mathrm{V}}],$$

where $W_i^{\mathrm{Q}}, W_i^{\mathrm{K}}, W_i^{\mathrm{V}} \in \mathbb{R}^{d_{\mathrm{model}} \times d_H}$ are the query, key, and value heads, and $d_H := d_{\mathrm{model}}/n_{\mathrm{H}}$ is the head dimension. Using a slightly refined definition of a head, which includes the output projection matrix $W_i^{\mathrm{O}} \in \mathbb{R}^{d_H \times d_{\mathrm{model}}}$, we have

$$\mathrm{head}_i = \mathrm{Attention}(XW_i^{\mathrm{Q}}, XW_i^{\mathrm{K}}, XW_i^{\mathrm{V}})W_i^{\mathrm{O}}$$
$$= \sigma \left( \frac{XW_i^{\mathrm{Q}}(XW_i^{\mathrm{K}})^{\top}}{\sqrt{d_H}} \right) XW_i^{\mathrm{V}}W_i^{\mathrm{O}}. \quad (1)$$

where $\sigma$ is the row-wise softmax activation.

Additionally, $\mathrm{head}_i \in \mathbb{R}^{N \times d_{\mathrm{model}}}$ which lets us rewrite the MHA formalism as

$$\mathrm{MHA}(X) = \sum_{i=1}^{n_{\mathrm{H}}} \mathrm{head}_i .$$

We propose to reformulate the attention formalism to allow for the construction of more memory efficient representations. Our derivation is based on two key observations:

First, the key and query matrices form a rank $d_H$ factorization of the *pre-softmax weight* $W_i := W_i^{\mathrm{Q}} W_i^{\mathrm{K},\top} \in \mathbb{R}^{d_{\mathrm{model}} \times d_{\mathrm{model}}}$ per head $i$. Analogously value and output matrices form a rank $d_H$ factorization of the *post-softmax weight* $\widetilde{W}_i := W_i^{\mathrm{V}} W_i^{\mathrm{O}} \in \mathbb{R}^{d_{\mathrm{model}} \times d_{\mathrm{model}}}$. Then, the MHA mechanism formally becomes

$$\mathrm{MHA}(X) = \sum_{i=1}^{n_{\mathrm{H}}} \sigma \left( \frac{XW_iX^{\top}}{\sqrt{d_H}} \right) X\widetilde{W}_i .$$

metric over parameter count.

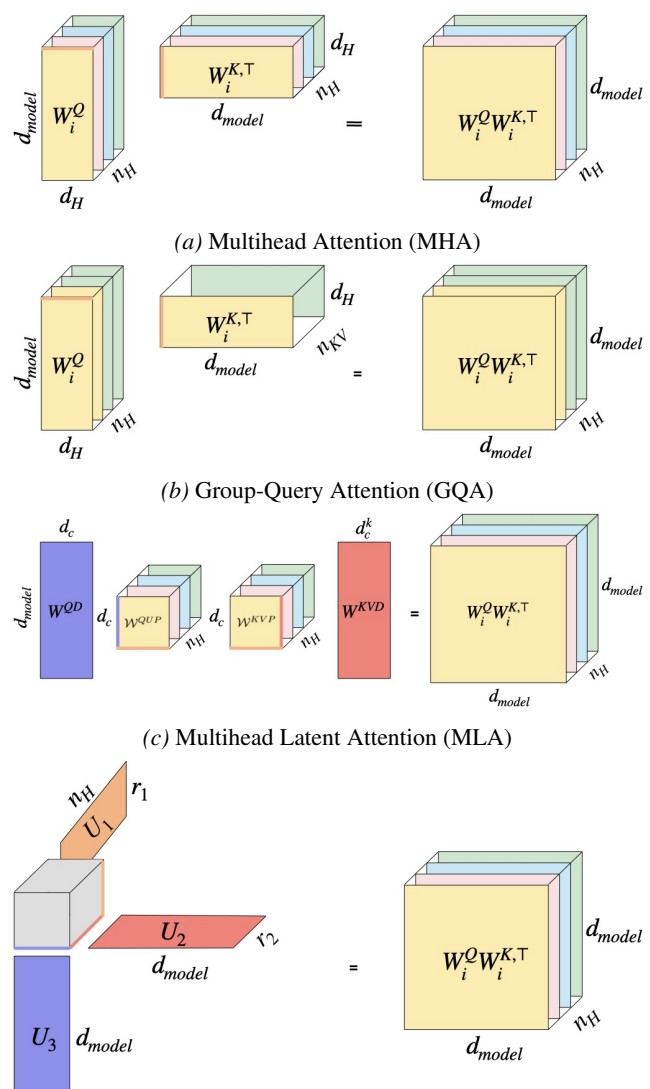

*(a)* Multihead Attention (MHA)

*(b)* Group-Query Attention (GQA)

*(c)* Multihead Latent Attention (MLA)

*(d)* Tucker Attention

*Figure 1.* Illustration of the pre-softmax attention tensor $\mathcal{W}$ under existing factorizations; colored edges denote contraction dimensions. **MHA:** $\mathcal{W}$ is formed by contracting $d_{\mathrm{model}} \times d_H$ query and key matrices along $d_H$ (orange) for each of the $n_{\mathrm{H}}$ heads, yielding tensor rank $(n_{\mathrm{H}}, d_{\mathrm{model}}, d_{\mathrm{model}})$. **GQA:** Queries follow the same parametrization as MHA, but only $n_{\mathrm{KV}}$ key matrices are used. During contraction along $d_H$ (orange), each key matrix is broadcasted to all queries in its head group, yielding rank $(n_{\mathrm{H}}, d_{\mathrm{model}}, n_{\mathrm{KV}}d_H)$. **MLA:** Key and query matrices are replaced by down- and up-projections, with up-projections reshaped to $n_{\mathrm{H}} \times d_{\mathrm{c}} \times d_H$. Contractions occur along $d_{\mathrm{c}}^{\mathrm{Q}}, d_{\mathrm{c}}^K$ (red/blue), and along $d_H$ per head, giving rank $(n_{\mathrm{H}}, d_{\mathrm{c}}^{\mathrm{Q}}, d_{\mathrm{c}}^K)$. At inference, contractions along $d_{\mathrm{c}}^{\mathrm{Q}}$ and $d_H$ are precomputed. The post-softmax tensor $\widetilde{\mathcal{W}}$ is parametrized analogously for MHA and GQA; for MLA, only the value matrix has low-rank factorization while the output matrix remains full-rank.

Thus, MHA is able to encode low-rank behavior only per-

head but not across heads.

Second, the *per-head* matrices $W_i$ and $\widetilde{W}_i$ can be collected as a tensor by stacking over all head indices, i.e.,

$$\mathcal{W} = (W_i)_{i=1}^{n_{\mathrm{H}}} \in \mathbb{R}^{n_{\mathrm{H}} \times d_{\mathrm{model}} \times d_{\mathrm{model}}}, \quad (2)$$

$$\widetilde{\mathcal{W}} = (\widetilde{W}_i^\top)_{i=1}^{n_{\mathrm{H}}} \in \mathbb{R}^{n_{\mathrm{H}} \times d_{\mathrm{model}} \times d_{\mathrm{model}}}. \quad (3)$$

For both $\mathcal{W}$ and $\widetilde{\mathcal{W}}$ the first mode is referred to as the head mode. For $\mathcal{W}$, the second and third modes correspond to the query and key computations. For $\widetilde{\mathcal{W}}$, the second and third modes correspond to the output and value computations. We stack the transpose of $\widetilde{W}_i$ in (3) for notational convenience; the key and value modes, which are often treated similarly, now align to the third mode.

To reformulate the MHA formula for these tensor-valued weights, we define

$$\mathcal{H}^{(1)}(X) := \sigma \left( \frac{\mathcal{W} \times_2 X \times_3 X}{\sqrt{d_{\mathrm{H}}}} \right) \in \mathbb{R}^{n_{\mathrm{H}} \times N \times N}, \quad (4)$$

$$\mathcal{H}^{(2)}(X) := \widetilde{\mathcal{W}} \times_3 X \in \mathbb{R}^{n_{\mathrm{H}} \times d_{\mathrm{model}} \times N}. \quad (5)$$

Here, $\mathcal{H}^{(1)}$ collects the $N \times N$ attention matrices across all heads, while $\mathcal{H}^{(2)}$ collects the projected value representations. The MHA output is then recovered by contracting these tensors over the head and sequence dimensions:

$$\mathrm{MHA}(X; \mathcal{W}, \widetilde{\mathcal{W}})_{jk} = \sum_{i,\ell} \mathcal{H}^{(1)}_{ij\ell}(X; \mathcal{W}) \mathcal{H}^{(2)}_{ik\ell}(X; \widetilde{\mathcal{W}}). \quad (6)$$

### 2.1. Tucker Attention

We now aim to find a formulation that enables us to further leverage shared information, e.g., when the column or row space of $W_i$ or $\widetilde{W}_i$ shares the same basis over different heads $i$. The natural framework for capturing such shared structure is the Tucker decomposition, which factorizes a tensor into a compact core and a set of basis matrices along each mode. This motivates the following definition.

**Definition 2.1** (Tucker Attention). Let $\mathcal{W} \in \mathbb{R}^{n_{\mathrm{H}} \times d_{\mathrm{model}} \times d_{\mathrm{model}}}$ and $\widetilde{\mathcal{W}} \in \mathbb{R}^{n_{\mathrm{H}} \times d_{\mathrm{model}} \times d_{\mathrm{model}}}$ denote the pre-softmax and post-softmax attention tensors from (2)–(3). *Tucker Attention* parametrizes these tensors via Tucker decompositions:

$$\mathcal{W} = \mathcal{C} \bigtimes_{j=1}^{3} U_j, \qquad \widetilde{\mathcal{W}} = \widetilde{\mathcal{C}} \bigtimes_{j=1}^{3} \widetilde{U}_j, \quad (7)$$

where $\mathcal{C} \in \mathbb{R}^{r_1 \times r_2 \times r_3}$ and $\widetilde{\mathcal{C}} \in \mathbb{R}^{\widetilde{r}_1 \times \widetilde{r}_2 \times \widetilde{r}_3}$ are learnable core tensors, and $U_1 \in \mathbb{R}^{n_{\mathrm{H}} \times r_1}$, $U_2 \in \mathbb{R}^{d_{\mathrm{model}} \times r_2}$, $U_3 \in \mathbb{R}^{d_{\mathrm{model}} \times r_3}$ (analogously for $\widetilde{U}_{1,2,3}$) are learnable basis matrices. The Tucker ranks $(r_1, r_2, r_3)$ and $(\widetilde{r}_1, \widetilde{r}_2, \widetilde{r}_3)$ control compression along the head, query/output, and key/value modes, respectively. Attention is computed via (6).

The basis matrices $U_2, U_3$ (resp. $\widetilde{U}_2, \widetilde{U}_3$) capture shared subspaces across heads in the query/key (resp. output/value computations), while $U_1$ and $\widetilde{U}_1$ enable compression across the head mode itself. This capability is absent in MHA, GQA, and MLA.

Tucker Attention is both *expressive* and *informative* to parameter efficient attention. We provide an overview to each of these properties below.

**Expressivity** Tucker Attention is expressive in that state-of-the-art attention methods such as MQA, GQA, and MLA are special cases of Tucker Attention with specific ranks. The key ingredient to this claim is showing, see Theorem B.2, that $\mathcal{W}$ in (2) admits the following Tucker factorization:

$$\mathcal{W} = \mathcal{C} \times_2 W^{\mathrm{Q}} \Pi \times_3 W^{\mathrm{K}} \Pi \quad (8)$$

where $\Pi$ is a column permutation matrix. Therefore, low-rank factorizations of $W^{\mathrm{Q}}$ and $W^{\mathrm{K}}$ directly infer the Tucker ranks of $\mathcal{W}$. Since MQA, GQA, and MLA all apply a specified low-rank factorization to all/some of the key $W^{\mathrm{K}}$, query $W^{\mathrm{Q}}$, and value $W^{\mathrm{V}}$ matrices, the Tucker ranks of $\mathcal{W}$ and $\widetilde{\mathcal{W}}$ for these methods are as follows:

- MHA – $\mathcal{W}$ & $\widetilde{\mathcal{W}}$: $(n_{\mathrm{H}}, d_{\mathrm{model}}, d_{\mathrm{model}})$
- MQA – $\mathcal{W}$ & $\widetilde{\mathcal{W}}$: $(n_{\mathrm{H}}, d_{\mathrm{model}}, d_{\mathrm{H}})$
- GQA – $\mathcal{W}$ & $\widetilde{\mathcal{W}}$: $(n_{\mathrm{H}}, d_{\mathrm{model}}, d_{\mathrm{H}} n_{\mathrm{KV}})$, where $n_{\mathrm{KV}}$ is the number of GQA KV heads.
- MLA – $\mathcal{W}$: $(n_{\mathrm{H}}, d_{\mathrm{c}}^{\mathrm{Q}}, d_{\mathrm{c}}^{\mathrm{K}})$, $\widetilde{\mathcal{W}}$: $(n_{\mathrm{H}}, d_{\mathrm{model}}, d_{\mathrm{c}}^{\mathrm{K}})$, where $d_{\mathrm{c}}^{\mathrm{K}}$ is the latent dimension for the key and value heads, and $d_{\mathrm{c}}^{\mathrm{Q}}$ is the latent dimension for the query heads.

Figure 1 illustrates the low-rank representations and contraction modes of the canonical approximate attention mechanisms, where MQA is the special case of GQA with $n_{\mathrm{KV}} = 1$ KV heads. Section 2.2 provides a brief overview of MQA, GQA, and MLA and a motivation for the above ranks. A more formal theory is presented in Appendix B.1.

**Informed Compression Across All Modes** By viewing parameter reduction through Tucker factorization, Tucker Attention enables compression along all tensor modes: head, query, key, value, and output. This exposes compression opportunities in the head and output modes that MQA, GQA, and MLA leave unexploited by construction.

To validate that such compressible structure exists in practice, we perform a *post-hoc* singular value analysis of the attention tensors $\mathcal{W}$ and $\widetilde{\mathcal{W}}$ from fully trained GPT-2 models (see Section 4.2 for training details). For each mode, we compute the matricization and plot the normalized singular spectrum in Figure 5 (Appendix), with the head mode shown in Figure 2.

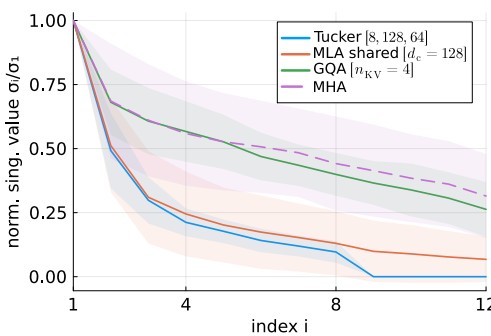

*Figure 2.* Normalized singular spectrum of pre-softmax head mode, $\mathrm{Mat}_1(\mathcal{W})$, for GPT2 pretraining. See Figure 5 for a detailed description. MLA shares a similar low-rank behavior across heads as Tucker Attention, but is not able to leverage this fact to reduce parameters.

The analysis reveals three key findings. First, MHA exhibits similar spectral decay across all modes (Figures 5c to 5f), yet GQA and MLA do not compress the output mode thereby retaining parameters that could be removed. Second, while GQA uses shared heads for the key and value projections, any low-rank behavior in the head mode is lost once the query and output projections are applied (see Figures 5a and 5b). Last, Figure 2 shows that MLA and Tucker Attention (with $r_1 = 8$) exhibit similar low-rank structure in the head mode, but MLA's parametrization cannot exploit this redundancy. Only through the Tucker lens can this head-mode compression be realized.

We claim that Tucker Attention realizes the low-rank tendencies shown above into quantifiable parameter reduction. In Table 1, we specify the parameter count for training and inference of MHA, MQA, GQA, MLA, and Tucker Attention. The inference costs are calculated post fusing components (for MLA) for additional savings. For a clear presentation, we only consider the case of Tucker Attention where the key, query, value, and output ranks are equal to a given rank $r$, and the head ranks pre- and post-softmax are $r_1$. The results show that the parameter count of all methods except Tucker Attention scale as $\mathcal{O}(d_{\mathrm{model}}^2)$ while Tucker Attention, by compressing in the output mode, scales as $\mathcal{O}(rd_{\mathrm{model}} + r_1 r^2)$. Therefore, if we assume $r \ll d_{\mathrm{model}}$, we expect savings in Tucker Attention over all other approaches without a significant decrease in validation metrics (see Section 4 for validation metric details). Figure 5 suggests that for GPT2 training, assumption $r \ll d_{\mathrm{model}}$ is true, and the results in Section 4 indicate that the low-rank ansatz does not harm performance.

## 2.2. MQA, GQA and MLA are special cases of Tucker Attention

In this section, we show that attention formulations like MQA, GQA, and MLA are compatible with Tucker Attention and a given Tucker rank. In this discussion, we only

*Table 1.* Comparison of the memory footprint of MHA, MQA, GQA with $n_{\mathrm{KV}}$ KV heads, MLA, and the proposed Tucker Attention. We count the total parameters of the key, query, value, and output weight matrices. MLA fuses query down and up projections during inference, thus the parameter count changes. We assume $d = n_{\mathrm{H}} d_{\mathrm{H}}$ for MHA, MQA, GQA, $n_{\mathrm{KV}}$ is the number of kv heads for GQA, $d_c$ is the latent dim/rank of MLA, and $(r_1, r, r)$ are the Tucker ranks of $\mathcal{W}$ and $\widetilde{\mathcal{W}}$ in Tucker Attention. Sequence length is denoted by $N$. Tucker Attention enjoys a favorable $\mathcal{O}(rd_{\mathrm{model}})$ scaling compared to the other methods.

| Method | #Params (train) | #Params (inference) | KV Cache |
|---|---|---|---|
| Eq.(6) | $4n_{\mathrm{H}}d_{\mathrm{model}}^2$ | $4d_{\mathrm{model}}^2$ | $4Nn_{\mathrm{H}}d_{\mathrm{model}}$ |
| MHA | $4d_{\mathrm{model}}^2$ | same as training | $2Nd_{\mathrm{model}}$ |
| MQA | $2d_{\mathrm{model}}^2 + 2d_{\mathrm{H}}d_{\mathrm{model}}$ | same as training | $2Nd_{\mathrm{H}}$ |
| GQA-$n_{\mathrm{KV}}$ | $2d_{\mathrm{model}}^2 + 2n_{\mathrm{KV}}d_{\mathrm{H}}d_{\mathrm{model}}$ | same as training | $2Nn_{\mathrm{KV}}d_{\mathrm{H}}$ |
| MLA (separated KV) | $d_{\mathrm{model}}^2 + 6d_{\mathrm{model}}d_c$ | $d_{\mathrm{model}}^2 + 4d_{\mathrm{model}}d_c$ | $2Nd_c$ |
| MLA (shared KV) | $d_{\mathrm{model}}^2 + 5d_{\mathrm{model}}d_c$ | $d_{\mathrm{model}}^2 + 2d_{\mathrm{model}}d_c$ | $Nd_c$ |
| Tucker (separated KV) | $2(n_{\mathrm{H}}r_1 + 2rd_{\mathrm{model}} + r_1 r^2)$ | same as training | $2Nr$ |
| Tucker (shared KV) | $2n_{\mathrm{H}}r_1 + 3rd_{\mathrm{model}} + 2r_1 r^2$ | same as training | $Nr$ |

focus on the ranks of the above methods with respect to pre-softmax compression, i.e., $\mathcal{W}$ in (2). The post-softmax ranks for $\widetilde{\mathcal{W}}$ and a more complete theory are found in Appendix B.1. Our discussion investigates the rank of $W^{\mathrm{Q}}$ and $W^{\mathrm{K}}$ of different attention mechanisms and concludes the Tucker ranks using Theorem B.2.

### 2.2.1. MULTI AND GROUP QUERY ATTENTION

Group query attention with selects $n_{\mathrm{KV}}$ K heads $W_i^{\mathrm{K}}$ (and analogously V heads), each of size $d_{\mathrm{model}} \times d_{\mathrm{H}}$, see Figure 1b. Therefore, $W^{\mathrm{K}}$ has a maximum rank of $n_{\mathrm{KV}}d_{\mathrm{H}}$. A similar shared compression technique is applied to the value heads. Thus the ranks in the key and value modes of $\mathcal{W}$ and $\widetilde{\mathcal{W}}$ are at most $n_{\mathrm{KV}}d_{\mathrm{H}}$. We view MQA as the special case of GQA with $n_{\mathrm{KV}} = 1$.

### 2.2.2. MULTIHEAD LATENT ATTENTION

In the case of MLA, the key heads are factorized as $W_i^{\mathrm{K}} = W^{\mathrm{DKV}}W_i^{\mathrm{UK}}$, where $W^{\mathrm{DKV}} \in \mathbb{R}^{d_{\mathrm{model}} \times d_c^{\mathrm{K}}}$ projects inputs into the latent space of dimension $d_c^{\mathrm{K}}$ and $W_i^{\mathrm{K}}$ prolongs the inputs up into the head-specific ambient space. Defining $W^{\mathrm{UK}} = [W_1^{\mathrm{UK}}, \cdots, W_{n_{\mathrm{H}}}^{\mathrm{UK}}] \in \mathbb{R}^{d_c^{\mathrm{Q}} \times d_{\mathrm{model}}}$, we have $W^{\mathrm{K}} = W^{\mathrm{DKV}}W^{\mathrm{UK}}$, and it is easy to see that $W^{\mathrm{K}}$ has a rank of $d_c^{\mathrm{K}}$. The query heads are factorized similarly to the key heads, yielding $W^{\mathrm{Q}} = W^{\mathrm{DQ}}W^{\mathrm{UQ}}$ where $W^{\mathrm{DQ}} \in \mathbb{R}^{d_{\mathrm{model}} \times d_c^{\mathrm{Q}}}$, and $W^{\mathrm{Q}}$ having a rank of $d_c^{\mathrm{Q}}$.

Stacking $W_i^{\mathrm{UK}}$, respectively $W_i^{\mathrm{UQ}}$, into tensors $\mathcal{W}^{\mathrm{UK}} \in \mathbb{R}^{n_{\mathrm{H}} \times d_{\mathrm{H}} \times d_c^{\mathrm{K}}}$ and $\mathcal{W}^{\mathrm{UQ}} \in \mathbb{R}^{n_{\mathrm{H}} \times d_{\mathrm{H}} \times d_c^{\mathrm{Q}}}$ yields the MLA formulation illustrated in Figure 1c. It is evident that the two up-projection tensors can be absorbed into the Tucker core of (8), yielding a more parameter-efficient representation, i.e., the Tucker Attention representation.

Lastly, we consider two versions of the value head compression. The first, *shared KV*, is the standard approach where

the down projection $W^{\mathrm{DKV}}$ is also used for value heads, i.e., $W_i^{\mathrm{V}} = W^{\mathrm{DKV}} W_i^{\mathrm{UV}}$. We also consider the case, called *separated KV*, where $W^{\mathrm{V}}$ uses its own down projection, i.e., $W_i^{\mathrm{V}} = W^{\mathrm{DV}} W_i^{\mathrm{UV}}$ where $W^{\mathrm{UV}} \in \mathbb{R}^{d_{\mathrm{model}} \times d_{\mathrm{c}}^{\mathrm{K}}}$.

# 3. Compatibility of Tucker Attention

In this section we show that Tucker Attention is compatible with KV caching, a modification of RoPE, and Flash-Attention.

## 3.1. KV Caching

To ease notation, we assume $r_3 = \widetilde{r}_3$. KV-caching can be realized in this formulation by computing and caching $K = XU_3 \in \mathbb{R}^{N \times r_3}$ and $V = X\widetilde{U}_3 \in \mathbb{R}^{N \times r_3}$. Further, we can update the attention matrix at the last token $x \in \mathbb{R}^{d_{\mathrm{model}}}$, i.e., the last row in the attention matrix, as

$$\sigma \left( \frac{\widetilde{\mathcal{C}} \times_1 U_1 \times_2 xU_2 \times_3 K}{\sqrt{d_{\mathrm{H}}}} \right) \times_3 V . \tag{9}$$

Caching $K$ and $V$ incurs a storage cost of $2Nr_3$. Additionally, Tucker Attention is compatible with a shared key-value down-projection by replacing $K, V$ with $L^{\mathrm{KV}} = XU_3 \in \mathbb{R}^{N \times r_3}$ in (9); further reducing the required KV-cache to $Nr_3$. For $r_3 = d_{\mathrm{c}}$ and shared KV projection, where $d_{\mathrm{c}}$ is the latent dimension in MLA, the cache size requirements between MLA and Tucker Attention are equal.

## 3.2. Rotary Position Embedding (RoPE)

A key question for practical adoption is whether Tucker Attention is compatible with rotary position embeddings (Su et al., 2023), the dominant position encoding in modern LLMs. Standard RoPE[2] applies rotation matrices $R(\ell, d) \in \mathbb{R}^{d \times d}$ to query and key vectors at token position $\ell$, where the rotation frequency depends on the embedding dimension $d$. The critical property is that the inner product between rotated queries and keys depends only on relative position:

$$R(m, d)R(n, d)^{\top} = R(m - n, d). \tag{10}$$

In standard MHA, RoPE is applied in the head dimension $d_{\mathrm{H}}$, yielding the pre-softmax entry at query position $m$ and key position $n$:

$$X_m W_i^{\mathrm{Q}} R(m, d_{\mathrm{H}}) R(n, d_{\mathrm{H}})^{\top} W_i^{\mathrm{K}, \top} X_n^{\top}. \tag{11}$$

Since Tucker Attention parametrizes the fused product $W_i^{\mathrm{Q}} W_i^{\mathrm{K}, \top}$ rather than individual query and key matrices, the standard placement of RoPE in the head dimension is incompatible. We resolve this by moving the rotation from the head dimension to the latent key dimension, yielding what we call *latent RoPE*.

---

[2]We refer to (Su et al., 2023), Equation 15 for the full definition.

**Definition 3.1** (Latent RoPE for Tucker Attention). Let $\hat{Q} = \mathcal{C} \times_1 U_1 \times_2 (XU_2) \in \mathbb{R}^{n_{\mathrm{H}} \times N \times r_3}$ and $K = XU_3 \in \mathbb{R}^{N \times r_3}$, with $\hat{Q}_m$ and $K_n$ denoting the representations at token positions $m$ and $n$. *Latent RoPE* computes the position-aware attention as:

$$\hat{Q}_m \times_3 \left( K_n R(n, r_3) R(m, r_3)^{\top} \right) \in \mathbb{R}^{n_{\mathrm{H}}}, \tag{12}$$

where $R(\ell, r_3) \in \mathbb{R}^{r_3 \times r_3}$ is the rotary embedding in the latent key space.

Latent RoPE satisfies the relative position property (10); see Appendix B.2 for the proof. The key insight is that rotational invariance holds regardless of where the rotation matrices are placed; our modification only changes the rotation frequency by operating in the latent dimension rather than the head dimension.

Importantly, latent RoPE is fully compatible with KV caching: the rotation matrices sit between the Tucker down-projector $U_3$ and the core $\mathcal{C}$, so the key rotation $R(n, r_3)$ is computed once and shared across all heads, reducing computational overhead compared to per-head RoPE in MHA and decoupled RoPE in MLA.

### 3.2.1. COROLLARY: SIMPLIFIED ROPE FOR MLA

A significant practical consequence of latent RoPE is that it provides a simpler alternative to the decoupled RoPE approach used in MLA (DeepSeek-AI et al., 2024). To our knowledge, this is the first demonstration that MLA is compatible with RoPE without requiring decoupled position encodings.

The original MLA implementation (DeepSeek-AI et al., 2024)[Section 2.1.3] introduces decoupled RoPE as a workaround: query and key matrices are partitioned into "semantic" and "rotational" components, with RoPE applied only to the rotational subset. This adds architectural complexity and prevents full weight fusion at inference time.

Since Tucker Attention generalizes MLA (Theorem B.4), latent RoPE applies directly to MLA in its native KV-latent space:

$$X_m W^{\mathrm{DQ}} W_i^{\mathrm{UQ}} W_i^{\mathrm{UV}} R(m, d_{\mathrm{c}}) R(n, d_{\mathrm{c}})^{\top} W^{\mathrm{DKV}, \top} X_n. \tag{13}$$

This formulation enables full inference-time fusion of the query-side matrices into a single projection $W_i^{\mathrm{Q,MLA}} := W^{\mathrm{DQ}} W_i^{\mathrm{UQ}} W_i^{UV} \in \mathbb{R}^{d_{\mathrm{model}} \times d_{\mathrm{c}}}$:

$$X_m W_i^{\mathrm{Q,MLA}} R(m, d_{\mathrm{c}}) R(n, d_{\mathrm{c}})^{\top} W^{\mathrm{DKV}, \top} X_n. \tag{14}$$

This simplification removes the need for separate semantic and rotational pathways, reducing MLA's implementation complexity while preserving KV-cache efficiency. We validate empirically in Table 6 that MLA is comparable with latent RoPE.

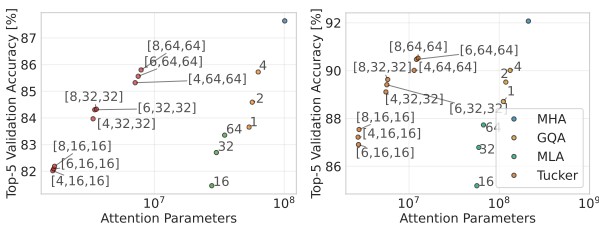

*(a)* ViT32l top-5 acc      *(b)* ViT14g top-5 acc

*Figure 3.* ViT32l (left) and ViT14g (right) top-5 validation performance on ImageNet1k for MHA and approximate attention mechanisms over total attention parameter count (top-left is best). GQA is presented with $n_{KV} = 1$ to $4$ KV heads, where GQA-1 corresponds to MQA. MLA is presented with latent dimension 16-64 with individual kv-weights. Tucker Attention is presented with ranks $[r_1, r_2, r_3]$. Is it apparent that Tucker Attention requires an order of magnitude fewer parameters for comparable accuracy levels and thus shifts the parameter-accuracy pareto-frontier to the top left.

### 3.3. Flash Attention

The proposed Tucker Attention is fully compatible with existing flash-attention implementations, requiring no custom kernels.

We briefly review the memory optimization in flash-attention (Dao, 2023). The key idea is to compute $\sigma\left(\frac{XW_i^Q (XW_i^K)^\top}{\sqrt{d_H}}\right) XW_i^V$ from (1) without materializing the $N \times N$ attention matrix, which would incur prohibitive memory costs for long sequences. Denoting $Q_i = XW_i^Q$, $K_i = XW_i^K$, and $V_i = XW_i^V$, flash-attention avoids this cost by partitioning $Q_i, K_i, V_i$ into sequence chunks of size $C \times d_H$ that fit in SRAM and accumulating the output via online softmax normalization; thus, the full attention matrix is never stored. The bottleneck then becomes how many times chunks must be loaded from HBM into SRAM.[3]

**Tucker attention unifies the advantages of MQA and MLA** Existing efficient attention methods offer complementary benefits for flash-attention: First, MQA and GQA reduce HBM accesses by sharing key-value heads across query heads, so only $n_H / n_{KV}$ KV chunks are loaded for all $n_H$ query heads. Second, MLA (in fused inference mode) reduces SRAM pressure by loading key and query chunks encoded in latent dimension $d_c^K$ instead of head dimension $d_H$, enabling larger chunk sizes.

Tucker Attention achieves both benefits simultaneously during training and inference: First, the key and value projections $K = XU_3 \in \mathbb{R}^{N \times r_3}$ and $V = X\widetilde{U}_3 \in \mathbb{R}^{N \times r_3}$ are

---

[3]SRAM (static random-access memory) refers to the small, fast on-chip cache on GPUs; HBM (high-bandwidth memory) is the larger but slower off-chip memory. On modern GPUs (e.g., NVIDIA H100), SRAM access is roughly 10–20× faster than HBM, making HBM-to-SRAM transfers the primary bottleneck in memory-bound operations like attention.

shared across all heads, so only a single KV chunk pair is loaded per attention computation for all query heads, just like MQA. Second, the chunks are encoded in the $r_3$ latent dimension, corresponding to the key and value ranks. Thus, just like MLA, the chunks loaded into SRAM induce less memory pressure than MHA or GQA chunks with encoding dimension $d_H$.

**Implementation** For evaluation, we precompute $K = XU_{r_3} \in \mathbb{R}^{N \times r_3}$, $V = X\widetilde{U}_{r_3} \in \mathbb{R}^{N \times r_3}$ (or load $K, V$ from the KV-cache), and compute the per-head queries $Q_i = \mathcal{C} \times_1 U_1 \times_2 (XU_2) \in \mathbb{R}^{N \times r_3}$. We then invoke the production-level flash-attention kernel in PyTorch, which natively supports GQA-style grouped heads. Since Tucker Attention uses a single shared KV pair for all query heads, this corresponds to GQA with $n_{KV} = 1$ (i.e., MQA) from the kernel's perspective.

We note that this evaluation strategy is one of several possible approaches; the Tucker structure may admit more efficient custom kernels that exploit the factorized core $\mathcal{C}$ directly. We defer exploration of specialized kernels to a future work.

## 4. Numerical Results

In this section, we compare MHA, GQA, MLA and the proposed Tucker Attention parametrization on vision transformers and GPT2. We use the notation for GQA and MLA presented in Section 2.2. For Tucker Attention, we fix $r_1 = \widetilde{r_1}$, $r_2 = \widetilde{r_2}$, and $r_3 = \widetilde{r_3}$, and present the ranks as $(r_1, r_2, r_3)$.

### 4.1. ViT16b on Cifar10/Cifar100 & ViT32l, ViT16g on ImageNet1k

**Setup** We initialize the model weights[4] from the ViT Huggingface endpoints and train on the corresponding downstream dataset. The low-rank parameterizations GQA, MLA and Tucker Attention are initialized by SVD-based approximation of the MHA weights; see Appendix C.1 for details, full training setup and additional tests.

**Findings** Figure 3 reports the mean of 5 training runs, with standard deviation of less than 0.1% per setup. We observe that the proposed Tucker Attention achieves a significantly better parameter-to-accuracy ratio than GQA and MLA: Tucker Attention requires almost an order of magnitude fewer trainable parameters to achieve similar performance than GQA and MLA. We further observe that lowering the ranks $r_2$ and $r_3$ is more beneficial for Tucker Attention than very low $r_1$ ranks, e.g. $r_1 \leq 2$.

---

[4]We refrain from full training due to the associated significant computational expense. Full pretraining results can be found in Section 4.2.

*Table 2.* Zero-shot results for GPT-2 [$d_{\text{model}} = 768, n_{\text{H}} = 12, d_{\text{H}} = 64$] variants with $N = 1024$ trained on OpenWebText with identical training hyperparameters, without fine-tuning. For each RoPE setting, we mark the and lowest KV cache and parameter storage (measured in MB and assuming BF16 encoding) of GQA, MLA, and Tucker Attention in bold. Tucker Attention achieves competitive validation accuracy and perplexity scores with a fewer trainable parameters and KV cache than the baseline methods.

| Method Metric | Attn params [MB] | KV Cache [MB] | Val loss (lower is better) | Perplexity (lower is better) | | | Accuracy [%] (higher is better) | | | | | | | | |
| | | | | WikiText2 | LAMBADA | Mean | LAMBADA | CBT-CN | CBT-NE | HellaSwag | WinoGrande | PIQA | ARC-E | ARC-C | Mean |
|---|---|---|---|---|---|---|---|---|---|---|---|---|---|---|---|
| MHA | 56.62 | 37.74 | 2.861 | 37.15 | 16.95 | 27.05 | 27.17 | 82.44 | 62.84 | 31.28 | 51.30 | 64.25 | 41.96 | 24.40 | 48.20 |
| GQA [$n_{\text{KV}} = 4$] | 37.74 | 12.85 | 2.896 | 42.97 | 22.30 | 32.63 | 25.13 | 82.24 | 61.52 | 30.77 | 51.78 | 63.17 | 42.17 | 25.94 | 47.84 |
| GQA [$n_{\text{KV}} = 2$] | 33.02 | 6.29 | 2.869 | 40.63 | 21.67 | 31.15 | 24.37 | 81.44 | 59.80 | 30.42 | 50.99 | 63.38 | 41.25 | 25.94 | 47.20 |
| MLA [GQA, $n_{\text{KV}} = 2$ equiv.] | 37.74 | 6.29 | 2.882 | 40.32 | 23.68 | 32.00 | 22.38 | 82.04 | 61.60 | 30.66 | 51.93 | 62.79 | 40.91 | 25.17 | 47.19 |
| MLA shared K,V [$d_c^{\text{K}}, d_c^{\text{Q}} = 128$] | 26.00 | **3.14** | 2.937 | 43.30 | 24.01 | 33.66 | 22.13 | 80.92 | 59.72 | 29.76 | 50.59 | 60.93 | 41.30 | 25.09 | 46.31 |
| Tucker [8, 128, 128] | 15.74 | 6.28 | 2.868 | 39.36 | 22.26 | 30.81 | 25.03 | 82.32 | 63.64 | 30.41 | 51.38 | 63.44 | 40.70 | 25.34 | 47.78 |
| Tucker [8,128,64] | 10.24 | **3.14** | 2.917 | 39.73 | 23.22 | 31.47 | 20.96 | 79.84 | 59.80 | 29.63 | 51.78 | 61.48 | 41.84 | 24.49 | 46.23 |
| Tucker [8,64,64] | **6.31** | **3.14** | 2.950 | 44.53 | 23.41 | 33.97 | 22.47 | 80.28 | 58.96 | 28.78 | 51.70 | 63.11 | 41.54 | 25.51 | 46.54 |
| MHA (Full RoPE) | 56.62 | 37.74 | 2.831 | 35.72 | 20.27 | 27.99 | 25.77 | 83.16 | 64.44 | 32.10 | 49.80 | 63.44 | 43.18 | 25.51 | 48.42 |
| GQA [$n_{\text{KV}} = 4$] | 37.74 | 12.85 | 2.851 | 36.35 | 19.02 | 27.69 | 26.35 | 83.20 | 63.20 | 31.83 | 52.96 | 63.71 | 43.69 | 26.11 | 48.80 |
| GQA [$n_{\text{KV}} = 2$] | 33.02 | 6.29 | 2.872 | 37.95 | 22.36 | 30.16 | 24.78 | 81.48 | 63.88 | 31.41 | 51.14 | 62.02 | 42.13 | 26.02 | 47.86 |
| MLA shared K,V [$d_c^{\text{K}}, d_c^{\text{Q}} = 128$] | 26.00 | **3.14** | 2.885 | 39.14 | 21.45 | 30.30 | 24.22 | 80.88 | 62.12 | 30.70 | 50.99 | 63.76 | 40.78 | 24.91 | 47.30 |
| Tucker [8, 128, 128] | 15.74 | 6.28 | 2.881 | 38.57 | 23.23 | 30.90 | 22.10 | 81.52 | 63.00 | 30.42 | 51.93 | 62.13 | 41.96 | 25.60 | 47.33 |
| Tucker [8,128,64] | 10.24 | **3.14** | 2.908 | 40.16 | 22.94 | 31.55 | 21.93 | 80.56 | 60.84 | 29.93 | 51.54 | 62.62 | 41.16 | 26.71 | 46.91 |
| Tucker [8, 64, 64] | **6.31** | **3.14** | 2.950 | 39.90 | 24.94 | 32.42 | 21.73 | 81.00 | 60.60 | 29.58 | 50.91 | 62.51 | 40.45 | 24.06 | 46.36 |

(Left margin labels: **w/o RoPE** for the first block, **w/ RoPE** for the second block.)

*Table 3.* Validation performance and per-iteration training timing breakdown for LLaMA3-1B variants on OpenWebText2 with $N = 4096$. Note LLaMA3 1B default setup is $n_{\text{KV}} = 8$. Training: Timing is reported per training iteration on 2 cluster nodes with 2 H100 GPUs each. Decode: We use query sequence length 1, with cached KV sequence length $N = 4096$. Timing is reported on 1 H100 GPU. Peak decode GPU memory load is measured in GB with batch size 1. All timings denote the mean over 10 iterations in [ms]. We remark that this implementation is non-optimized research code. Tucker Attention achieves competitive validation loss and perplexity scores at significant parameter and latency reductions in training and decoding mode.

| Method | Attn Params [MB] | Eval | | Training Efficiency | | | | Decode Efficiency | | |
| | | PPL ↓ | Val loss ↓ | Fwd [ms] ↓ | Bwd [ms] ↓ | Opt [ms] ↓ | Train Iter [ms] ↓ | KV Cache [MB] | Peak Mem [GB] | Latency [ms] ↓ |
|---|---|---|---|---|---|---|---|---|---|---|
| MHA | 268 | 12.87 | 2.55 | 1.58 | 3.51 | 0.101 | 5.19 | 268 | 6.5 | 1.379 |
| GQA [$n_{\text{KV}} = 8$] | 168 | 13.62 | 2.61 | 1.47 | 3.30 | 0.087 | 4.87 | 67 | 5.7 | 1.310 |
| GQA [$n_{\text{KV}} = 2$] | 142 | 13.87 | 2.63 | 1.44 | 3.20 | 0.086 | 4.73 | 16.7 | 5.7 | 1.305 |
| MQA [$n_{\text{KV}} = 1$] | 138 | 14.27 | 2.64 | 1.43 | 3.13 | 0.083 | 4.67 | 8.38 | 5.6 | 1.296 |
| MLA [$d_c^{\text{K}}, d_c^{\text{Q}} = 128$] | 93.3 | 14.54 | 2.68 | 1.37 | 2.96 | 0.079 | 4.42 | 16.7 | 5.9 | 1.341 |
| MLA [$d_c^{\text{K}}, d_c^{\text{Q}} = 64$] | 79.2 | 14.77 | 2.69 | 1.34 | 2.94 | 0.076 | 4.36 | 8.38 | 5.6 | 1.317 |
| MLA [$d_c^{\text{K}}, d_c^{\text{Q}} = 32$] | 73.4 | 15.65 | 2.75 | 1.33 | 2.88 | 0.075 | 4.30 | 4.19 | 5.4 | 1.292 |
| Tucker [32, 128, 128] | 33.5 | 14.29 | 2.65 | 1.40 | 3.02 | 0.071 | 4.41 | 16.7 | 5.4 | 1.306 |
| Tucker [32, 128, 64] | 21.0 | 14.60 | 2.68 | 1.35 | 2.89 | 0.069 | 4.33 | 8.38 | 5.2 | 1.294 |
| Tucker [32, 64, 64] | 12.6 | 14.76 | 2.69 | 1.30 | 2.70 | 0.067 | 4.05 | 8.38 | 5.2 | 1.274 |

## 4.2. GPT2 on OpenWebText

**Setup** We compare MHA, GQA, MLA and the proposed Tucker Attention attention parametrizations on GPT2. We initialize the model from scratch with random weights, fixing the random seed for comparability between methods. We use the same training hyperparameters for all methods, see Appendix C.2 for details. For each attention mechanism we train GPT2 twice: once equipped with a position embedding vector in the embedding layer and once using RoPE, where we remove the position embedding vector and instead apply RoPE at each pre-softmax attention layer. **MHA and GQA setup:** Both methods are implemented canonically. **MLA setup:** Unless noted otherwise, we follow the implementation of (DeepSeek-AI et al., 2024) for consistency. Thus we train MLA in *unfused mode* with *decoupled RoPE*. **Tucker Attention setup:** We use the parametrization of Equation (7), and the proposed variation of Section 3.2.

**Findings** Figure 4 (Appendix) reports the validation loss curve[5] of GPT2 with classical position embedding for MHA, GQA ($n_{\text{KV}} = 2$), MLA ($d_c^{\text{Q}} = d_c^{\text{K}} = 128$) with

---

[5]The training loss curve follows the same structure.

shared KV down-projection, and Tucker Attention with ranks $(8, 128, 64)$. In this case, MLA and Tucker Attention have the same KV cache size during inference. We observe that Tucker Attention converges at the same rate as the canonical methods, reaching a lower loss than MLA. Throughout the GPT2 experiments we observed that MLA with shared KV exhibits small instabilities, and even diverges for $d_c^{\text{K}} = d_c^{\text{Q}} \leq 128$.

Table 2 compares the GPT2 variants' validation performance on a set of one-shot validation tests, reporting perplexity (lower is better), validation loss (lower is better), and validation accuracy (higher is better), as well as the total number of trainable parameters in the attention layers and the total required KV cache during inference. In the category w/o RoPE, we observe that Tucker Attention with ranks $(8, 128, 64)$ requires about 18% of the parameters of MHA and 39% of the MLA parameters to obtain competitive validation results. That is, Tucker Attention achieves the similar mean perplexity and similar mean accuracy as the baseline methods.

Similarly, we observe in the second part of the table that Tucker Attention is compatible with latent RoPE, imple-

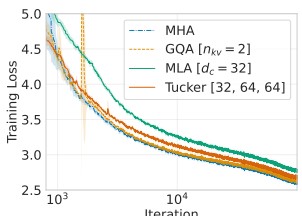

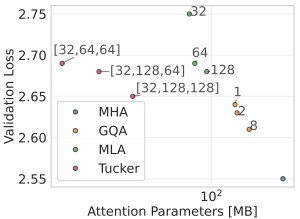

*(a)* Training loss history. Tucker Attention converges as well as the baseline methods.

*(b)* Validation loss plotted against parameter count. Tucker Attention pushes the Pareto frontier to the lower left.

*Figure 4.* Llama3-1B training cross-entropy loss (left) and final time validation cross-entropy loss (right) on OpenWebtext2 with RoPE in Bf16 floating point accuracy. Approximate attention methods are MLA GQA, and Tucker Attention. Tucker Attention converges well, despite having only a fraction of the trainable parameters of the other methods, and does not exhibits instabilities.

mented using Equation (12). The perplexity and test accuracy values of GPT-2 are slightly improved compared to the models without RoPE, mirroring the modest gain in test scores of MLA, GQA, and MHA.

Next, we validate the insights of Section 2.1 on expressivity of GQA and MLA. In particular, MLA with $d_c^Q = 768$, $d_c^K = 128$, and a separate KV down projection expresses the same ranks of $\mathcal{W}$ as GQA with $n_{KV} = 2$. Since the query weights are full-rank, we fuse $W^{DQ}W_i^{UQ}W_i^{UV}$ into one matrix for training stability. We refer to this MLA setup as "MLA [GQA, $n_{KV} = 2$ equiv.]" in Table 2, and observe that both the mean perplexity and mean validation accuracy are comparable to GQA $n_{KV} = 2$, confirming the expressivity hypothesis.

### 4.3. GPT2 on Shakespeare

**Setup** We repeat the setup of Section 4.2, using the smaller Shakespeare dataset and reduced sequence length of $N = 256$, see Appendix C.2 for details.

**Findings** We observe in Table 6 that Tucker Attention achieves a comparable validation loss values using over an order of magnitude fewer attention parameters and KV cache than the baseline methods, including MLA. Further, we observe that shared KV down-projection is compatible with Tucker Attention at a slight increase in validation accuracy, which is expected due to the further reduction in trainable parameters. This effect is also observable when comparing MLA with separate and shared KV down-projection. Validating Corollary Section 3.2.1, we observe that MLA is compatible with latent RoPE.

### 4.4. LLaMA3-1B on OpenWebText2

**Setup** We train a 1 Billion parameter LLaMA 3-style decoder-only transformer from scratch on OpenWeb-Text2 (Gao et al., 2020) using GPT-NeoX (Andonian et al., 2023). The baseline attention mechanism is GQA with $n_H = 32$ query heads, with head dimension $d_H = 64$ and

$n_{KV} = 8$ KV heads, and sequence length $N = 4096$. See C.3 for full details on training hyperparameters. We compare to MHA, MQA, and Tucker Attention in Table 3, where each setup is run once using a fixed random seed. Across methods, all non-attention components (embeddings, MLP, norms, RoPE settings) and training hyperparameters are kept fixed. GPT-NeoX[6] serves as the framework for the test, and we use the built-in timers of the megatron backend to measure wall-time cost. Reported timings reflect end-to-end evaluation, including uncompressed MLP layers. We remark that the Tucker Attentionlayer is non-optimized research code and the timing results should be treated as a general indicated rather than a definite roofline. Training is facilitated on 2 cluster nodes with 2 Nvidia H100 each, and the training timings are specific to this setup. Inference decoding is facilitated on one H100 GPU with $N = 4096$ cached KV tokens, 1 query token, and batch size 1.

**Findings** A similar trend appears as in the GPT-2 results: Tucker Attention uses only 10–20% of the LLaMA3 baseline (GQA, $n_{KV} = 8$), with a modest 1–3% increase in validation cross-entropy loss. It reduces the KV cache by 4–8×, depending on rank. This reduction in parameters and KV cache lowers training cost, decreasing iteration wall time by up to 20% vs. MHA and 15% vs. the GQA ($n_{KV} = 8$) baseline. Additionally, Tucker Attention outperforms MLA in both validation loss and parameter count.

**We conclude** that the introduced viewpoint of treating the pre- and post-softmax weights as one tensor-valued object each yields several practical benefits. The deduced Tucker Attention scheme is more parameter efficient than existing methods, allows introducing latent RoPE, is compatible with KV caching, and combines the memory benefits of MLA and GQA for FlashAttention during training and inference.

## Acknowledgments

This manuscript has been authored by UT-Battelle, LLC under Contract No. DE-AC05-00OR22725 with the U.S. Department of Energy. The United States Government retains and the publisher, by accepting the article for publication, acknowledges that the United States Government retains a non-exclusive, paid-up, irrevocable, world-wide license to publish or reproduce the published form of this manuscript, or allow others to do so, for United States Government purposes. The Department of Energy will provide public access to these results of federally sponsored research in accordance with the DOE Public Access Plan(http://energy.gov/downloads/doe-public-access-plan).

---

[6]We use a fork of Eleuther AI's github repository https://github.com/eleutherai/gpt-neox.

S. Schotthöfer and S. Schnake were supported by the Laboratory Directed Research and Development Program of Oak Ridge National Laboratory (ORNL), managed by UT-Battelle, LLC for the U.S. Department of Energy.

This research used resources of the Experimental Computing Laboratory (ExCL) at the Oak Ridge National Laboratory, which is supported by the Office of Science of the U.S. Department of Energy under Contract No. DE-AC05-00OR22725.

T. Klein and S. Sager received funding from the German Federal Joint Committee (Grant 01VSF23017), from the European Regional Development Fund (Grants Timing Matters and IntelAlgen) under the European Union's Horizon Europe Research and Innovation Program, and from the German Research Foundation DFG (Grant GRK 2297), which we gratefully acknowledge.

## Impact Statement

This paper presents work whose goal is to advance the field of parameter efficient Machine Learning. There are many potential societal consequences of our work, none which we feel must be specifically highlighted here.

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

# A. Notation Overview

We summarize the notation used throughout this paper in Table 4.

*Table 4.* Summary of notation.

| Symbol | Description |
|---|---|
| *General conventions* | |
| $x, y$ | Scalars and vectors (lowercase italic) |
| $W, X$ | Matrices (uppercase italic) |
| $\mathcal{C}, \mathcal{W}$ | Tensors (uppercase calligraphic) |
| *Dimensions* | |
| $N$ | Sequence length (number of tokens) |
| $d_{\text{model}}$ | Model embedding dimension |
| $n_{\text{H}}$ | Number of attention heads |
| $d_{\text{H}}$ | Per-head dimension ($d_{\text{H}} = d_{\text{model}}/n_{\text{H}}$) |
| $n_{\text{KV}}$ | Number of KV heads in GQA |
| $d_{\text{c}}, d_{\text{c}}^{\text{Q}}, d_{\text{c}}^{\text{K}}$ | Latent dimensions in MLA |
| *Input and attention weights* | |
| $X \in \mathbb{R}^{N \times d_{\text{model}}}$ | Embedded input sequence |
| $W^{\text{Q}}, W^{\text{K}}, W^{\text{V}} \in \mathbb{R}^{d_{\text{model}} \times d_{\text{model}}}$ | Query, key, value weight matrices |
| $W_i^{\text{Q}}, W_i^{\text{K}}, W_i^{\text{V}} \in \mathbb{R}^{d_{\text{model}} \times d_{\text{H}}}$ | Per-head query, key, value weights |
| $W_i^{\text{O}} \in \mathbb{R}^{d_{\text{H}} \times d_{\text{model}}}$ | Per-head output projection |
| $W_i = W_i^{\text{Q}} W_i^{\text{K}\top}$ | Pre-softmax weight for head $i$ |
| $\widetilde{W}_i = W_i^{\text{V}} W_i^{\text{O}}$ | Post-softmax weight for head $i$ |
| *Attention tensors* | |
| $\mathcal{W} \in \mathbb{R}^{n_{\text{H}} \times d_{\text{model}} \times d_{\text{model}}}$ | Pre-softmax attention tensor |
| $\widetilde{\mathcal{W}} \in \mathbb{R}^{n_{\text{H}} \times d_{\text{model}} \times d_{\text{model}}}$ | Post-softmax attention tensor |
| *Tucker decomposition* | |
| $\mathcal{C} \in \mathbb{R}^{r_1 \times r_2 \times r_3}$ | Core tensor for $\mathcal{W}$ |
| $\widetilde{\mathcal{C}} \in \mathbb{R}^{\tilde{r}_1 \times \tilde{r}_2 \times \tilde{r}_3}$ | Core tensor for $\widetilde{\mathcal{W}}$ |
| $(r_1, r_2, r_3)$ | Tucker ranks of $\mathcal{W}$ (head, query, key modes) |
| $(\tilde{r}_1, \tilde{r}_2, \tilde{r}_3)$ | Tucker ranks of $\widetilde{\mathcal{W}}$ (head, output, value modes) |
| $U_1 \in \mathbb{R}^{n_{\text{H}} \times r_1}$ | Basis matrix for head mode of $\mathcal{W}$ |
| $U_2 \in \mathbb{R}^{d_{\text{model}} \times r_2}$ | Basis matrix for query mode of $\mathcal{W}$ |
| $U_3 \in \mathbb{R}^{d_{\text{model}} \times r_3}$ | Basis matrix for key mode of $\mathcal{W}$ |
| $\widetilde{U}_1 \in \mathbb{R}^{n_{\text{H}} \times \tilde{r}_1}$ | Basis matrix for (post-softmax) head mode of $\widetilde{\mathcal{W}}$ |
| $\widetilde{U}_2 \in \mathbb{R}^{d_{\text{model}} \times \tilde{r}_2}$ | Basis matrix for output mode of $\widetilde{\mathcal{W}}$ |
| $\widetilde{U}_3 \in \mathbb{R}^{d_{\text{model}} \times \tilde{r}_3}$ | Basis matrix for value mode of $\widetilde{\mathcal{W}}$ |
| *MLA-specific notation* | |
| $W^{\text{DKV}} \in \mathbb{R}^{d_{\text{model}} \times d_{\text{c}}^{\text{K}}}$ | Down-projection for keys/values |
| $W^{\text{DQ}} \in \mathbb{R}^{d_{\text{model}} \times d_{\text{c}}^{\text{Q}}}$ | Down-projection for queries |
| $W^{\text{UK}}, W^{\text{UQ}}, W^{\text{UV}}$ | Up-projection matrices |
| *Tensor operations* | |
| $\text{Mat}_j(\mathcal{T})$ | Mode-$j$ matricization (unfolding) of tensor $\mathcal{T}$ |
| $\mathcal{T} \times_j M$ | Mode-$j$ product of tensor $\mathcal{T}$ with matrix $M$ |
| $\bigtimes_{j=1}^{3}$ | Shorthand for successive mode products |
| *Other notation* | |
| $\sigma(\cdot)$ | Row-wise softmax activation |
| $R(\ell, d) \in \mathbb{R}^{d \times d}$ | RoPE rotation matrix at position $\ell$ |
| $\text{col}_i(Z)$ | The $i$-th column of matrix $Z$ |

# B. Proofs

## B.1. Tucker Representations of MQA, GQA, and MLA

Here we give a formal treatment for the ranks of MQA, GQA, and MLA given in Section 2.1. We first give a technical lemma before showing (8), which is included in Theorem B.2.

**Lemma B.1.** *Assume $D_i = A_i B_i^\top$, where $A_i, B_i \in \mathbb{R}^{d_{\mathrm{model}} \times d_{\mathrm{H}}}$, and $D_i \in \mathbb{R}^{d_{\mathrm{model}} \times d_{\mathrm{model}}}$ are matrices. Further, let $\mathcal{A}, \mathcal{B} \in \mathbb{R}^{n_{\mathrm{H}} \times d_{\mathrm{model}} \times d_{\mathrm{H}}}$, and $\mathcal{D} \in \mathbb{R}^{n_{\mathrm{H}} \times d_{\mathrm{model}} \times d_{\mathrm{model}}}$ be the corresponding tensors with $\mathcal{A}_{ijk} = (A_i)_{jk}$ for all $i = 1, \ldots, n_{\mathrm{H}}$. Then, $\mathcal{D}$ can be written as a tensor $\mathcal{D} = \mathcal{C} \times_2 \mathrm{Mat}_2(\mathcal{A}) \times_3 \mathrm{Mat}_2(\mathcal{B})$ with $\mathcal{C} \in \mathbb{R}^{n_{\mathrm{H}} \times d_{\mathrm{model}} \times d_{\mathrm{model}}}$. More precisely, denoting the vectorized index $(i\ell) := (i-1) \cdot d_{\mathrm{H}} + \ell \in \{1, \cdots, d_{\mathrm{model}}\}$, we have*

$$\mathcal{C}_{inm} = \sum_{\ell=1}^{d_{\mathrm{H}}} \delta_{n,(i\ell)} \delta_{m,(i\ell)} \,.$$

*Proof.* The result is immediate from the following calculation:

$$
\begin{aligned}
\mathcal{D}_{ijk} &= \sum_{\ell=1}^{d_{\mathrm{H}}} \mathcal{A}_{ij\ell} \mathcal{B}_{ik\ell} \\
&= \sum_{\ell=1}^{d_{\mathrm{H}}} \mathrm{Mat}_2(\mathcal{A})_{j,(i\ell)} \mathrm{Mat}_2(\mathcal{B})_{k,(i\ell)} \\
&= \sum_{m,n=1}^{d_{\mathrm{model}}} \sum_{\ell=1}^{d_{\mathrm{H}}} \mathrm{Mat}_2(\mathcal{A})_{j,n} \delta_{n,(i\ell)} \delta_{m,(i\ell)} \mathrm{Mat}_2(\mathcal{B})_{k,m} = \sum_{m,n=1}^{d_{\mathrm{model}}} \mathcal{C}_{inm} \mathrm{Mat}_2(\mathcal{A})_{j,n} \mathrm{Mat}_2(\mathcal{B})_{k,m} \,.
\end{aligned}
$$

$\square$

**Theorem B.2.** *Assume $W_i = W_i^{\mathrm{Q}} W_i^{\mathrm{K},\top}$, where $W_i^{\mathrm{Q}}, W_i^{\mathrm{K}} \in \mathbb{R}^{d_{\mathrm{model}} \times d_{\mathrm{H}}}$, and $W_i \in \mathbb{R}^{d_{\mathrm{model}} \times d_{\mathrm{model}}}$ are matrices. Then the pre-softmax tensor $\mathcal{W} \in \mathbb{R}^{n_{\mathrm{H}} \times d_{\mathrm{model}} \times d_{\mathrm{model}}}$ with $\mathcal{W}_{ijk} = (W_i)_{jk}$ admits the Tucker decomposition in (8) and has a maximal Tucker rank of $(n_{\mathrm{H}}, \mathrm{rank}(W^{\mathrm{Q}}), \mathrm{rank}(W^{\mathrm{K}}))$.*

*Proof.* Define $\mathcal{A}, \mathcal{B} \in \mathbb{R}^{n_{\mathrm{H}} \times d_{\mathrm{H}} \times d_{\mathrm{model}}}$ by $\mathcal{A}_{ijk} = (W_i^{\mathrm{Q}})_{jk}$ and $\mathcal{A}_{ijk} = (W_i^{\mathrm{Q}})_{jk}$, where we choose the tensorization such that $\mathcal{A}_{ijk} = (W_i^{\mathrm{Q}})_{jk}$ and $\mathcal{B}_{ijk} = (W_i^{\mathrm{Q}})_{jk}$. Note that $\mathrm{Mat}_2(\mathcal{A})$ and $\mathrm{Mat}_2(\mathcal{B})$ are column permutations of $W^{\mathrm{Q}}$ and $W^{\mathrm{K}}$ respectively. Then applying Lemma B.1, with $\mathcal{D} = \mathcal{W}$, yields

$$\mathcal{W} = \mathcal{C} \times_2 \mathrm{Mat}_2(\mathcal{A}) \times_3 \mathrm{Mat}_2(\mathcal{B}) = \mathcal{C} \times_2 W^{\mathrm{Q}} \Pi \times_3 W^{\mathrm{K}} \Pi,$$

where $\Pi$ is a permutation matrix. Hence (8) is immediate. Let $r_{\mathrm{Q}} := \mathrm{rank}(W^{\mathrm{Q}})$, then we can write $W^{\mathrm{Q}} = V^{\mathrm{Q}} C^{\mathrm{Q}}$ with $V^{\mathrm{Q}} \in \mathbb{R}^{d_{\mathrm{model}} \times r_{\mathrm{Q}}}$ and $C^{\mathrm{Q}} \in \mathbb{R}^{r_{\mathrm{Q}} \times d_{\mathrm{model}}}$. In the same manner, we have $W^{\mathrm{K}} = V^{\mathrm{K}} C^{\mathrm{K}}$ with $V^{\mathrm{K}} \in \mathbb{R}^{d_{\mathrm{model}} \times r_{\mathrm{K}}}$, $C^{\mathrm{K}} \in \mathbb{R}^{r_{\mathrm{K}} \times d_{\mathrm{model}}}$, and $r_{\mathrm{K}} := \mathrm{rank}(W^{\mathrm{K}})$. Thus,

$$\mathcal{W} = \mathcal{C} \times_2 V^{\mathrm{Q}} C^{\mathrm{Q}} \times_3 V^{K} C^{K} = (\mathcal{C} \times_2 C^{\mathrm{Q}} \times_3 C^{K}) \times_2 V^{\mathrm{Q}} \times_3 V^{K},$$

where $\widehat{\mathcal{C}} := \mathcal{C} \times_2 C^{\mathrm{Q}} \times_3 C^{K}$ is an element of $\mathbb{R}^{n_{\mathrm{H}} \times r_{\mathrm{Q}} \times r_{\mathrm{K}}}$. Therefore, the Tucker rank of $W$ is at most $(n_{\mathrm{H}}, \mathrm{rank}(W^{\mathrm{Q}}), \mathrm{rank}(W^{\mathrm{K}}))$. $\square$

We now show the maximal Tucker ranks for GQA and MLA as presented in (2.2). We note that MQA is a special case of GQA with $n_{\mathrm{KV}} = 1$.

**Theorem B.3.** *The pre-softmax attention tensor $\mathcal{W}$ and post-softmax attention tensor $\widetilde{\mathcal{W}}$ for Group Query Attention with $n_{\mathrm{KV}}$ KV heads both have maximal Tucker rank $(n_{\mathrm{H}}, d_{\mathrm{model}}, n_{\mathrm{KV}} d_{\mathrm{H}})$.*

*Proof.* Let $W_g^{\mathrm{K,GQA}} \in \mathbb{R}^{d_{\mathrm{model}} \times d_{\mathrm{H}}}$ for $g = 1, \ldots, n_{\mathrm{KV}}$ be the number of key heads. Then the GQA key heads satisfy $W_h^{\mathrm{K}} \in \{W_g^{\mathrm{K,GQA}}\}_{g=1}^{n_{\mathrm{KV}}}$ for every $h = 1, \ldots, n_{\mathrm{H}}$. Hence, $\mathrm{span}(W^{\mathrm{Q}}) = \mathrm{span}([W_1^{\mathrm{K,GQA}}, \ldots, W_{n_{\mathrm{KV}}}^{\mathrm{K,GQA}}])$. Since $[W_1^{\mathrm{K,GQA}}, \ldots, W_{n_{\mathrm{KV}}}^{\mathrm{K,GQA}}] \in \mathbb{R}^{d_{\mathrm{model}} \times d_{\mathrm{H}} n_{\mathrm{KV}}}$, its rank and the rank of $W^{\mathrm{Q}}$ must be at most $d_{\mathrm{H}} n_{\mathrm{KV}}$. Therefore by Theorem B.2, $r_3 \leq d_{\mathrm{H}} n_{\mathrm{KV}}$.

The same argument holds for $\widetilde{\mathcal{W}}$ with $W^{\mathrm{Q}}$ and $W^{\mathrm{K}}$ swapped for $W^{\mathrm{O}}$ and $W^{\mathrm{V}}$ respectively. $\square$

**Theorem B.4.** *The pre-softmax attention tensor $\mathcal{W}$ for Multihead Latent Attention with latent query dimension $d_c^Q$ and latent key dimension $d_c^K$ has maximal Tucker rank $(n_H, d_c^Q, d_c^K)$. The post-softmax attention tensor $\widetilde{\mathcal{W}}$ with latent value dimension $d_c^K$ has maximal Tucker rank $(n_H, d_{\text{model}}, d_c^K)$.*

*Proof.* In MLA, $W^Q = W^{DKV}W^{UK}$ where $W^{DKV} \in \mathbb{R}^{d_{\text{model}} \times d_c^K}$. Hence $\text{rank}\, W^Q \leq d_c^K$, and, by Theorem B.2, $r_3 \leq d_c^K$. A similar argument shows $r_2 \leq d_c^Q$.

The same argument can be repeated for $\widetilde{\mathcal{W}}$ with $W^V = W^{DKV}W^{UV}$ and $W^O$ full-rank. $\qquad\qquad\qquad\qquad\square$

### B.2. Rotary Position Embedding

**Lemma B.5.** *The expression of* (12) *fulfills the condition of* (10).

*Proof.* It is straightforward to see that for the token positions $m, n$

$$
\begin{aligned}
\hat{Q}_m \times_3 \left(K_n R(n, r_3) R(m, r_3)^\top\right) \in \mathbb{R}^{n_H} &= \left(\hat{Q}_m \times_3 R(m, r_3)^\top\right) \times_3 \left(K_n R(n, r_3)\right) \\
&= \sum_{\ell_3} \left(\sum_{\ell_1} \hat{Q}_{\cdot,m,\ell_1} R(m, r_3)_{\ell_1, \ell_3}\right) \left(\sum_{\ell_2} K_{n,\ell_2} R(n, r_3)_{\ell_2, \ell_3}\right) \\
&= \sum_{\ell_3} \sum_{\ell_2} \sum_{\ell_1} \left(\hat{Q}_{\cdot,m,\ell_1} R(m, r_3)_{\ell_1, \ell_3} K_{n,\ell_2} R(n, r_3)_{\ell_2, \ell_3}\right) \\
&= \sum_{\ell_2} \sum_{\ell_1} \left(\hat{Q}_{\cdot,m,\ell_1} \sum_{\ell_3} \left(R(m, r_3)_{\ell_1, \ell_3} R(n, r_3)_{\ell_3, \ell_2}^\top\right) K_{n,\ell_2}\right) \\
&= \sum_{\ell_2} \sum_{\ell_1} \left(\hat{Q}_{\cdot,m,\ell_1} R(m - n, r_3)_{\ell_1, \ell_2} K_{n,\ell_2}\right) \\
&= \left(\hat{Q}_m \times_3 R(m - n, r_3)^\top\right) \times_3 K_n
\end{aligned}
\tag{15}
$$

$\qquad\qquad\qquad\qquad\qquad\qquad\qquad\qquad\qquad\qquad\qquad\qquad\qquad\qquad\qquad\qquad\qquad\square$

## C. Details to the numerical experiments

### C.1. Transfer Learning with Vision Transformers

#### C.1.1. CIFAR-10/100 DATA

The CIFAR-10 dataset comprises 60,000 RGB images of size $32 \times 32$ pixels, uniformly distributed across 10 object classes. We apply standard data augmentation techniques to the training set, including random horizontal flipping followed by normalization with mean $[0.4914, 0.4822, 0.4465]$ and standard deviation $[0.2470, 0.2435, 0.2616]$. The test set is only normalized. The same augmentation strategy is applied to CIFAR-100, using mean $[0.5071, 0.4867, 0.4408]$ and standard deviation $[0.2673, 0.2564, 0.2762]$. ViT models receive images resized to $224 \times 224$ through the data pipeline.

#### C.1.2. IMAGENET-1K DATA

The ImageNet dataset consists of 1000 classes and over 1.2 million RGB training images, with a standard resolution of $224 \times 224$ pixels. We follow the standard data augmentation pipeline for ImageNet, which includes a random resized crop to $224 \times 224$, and normalization using mean $[0.5, 0.5, 0.5]$ and standard deviation $[0.5, 0.5, 0.5]$. The test set is only resized and center-cropped to $224 \times 224$, followed by normalization.

#### C.1.3. VIT MODELS

In this paper, we use a Pytorch implementation for neural network training. We take pretrained weights from the Imagenet1k dataset as initialization. The data-loader randomly samples a batch for each batch-update which is the only source of randomness in our training setup. Below is an overview of the used network architectures

- ViT-B.16 is a Vision Transformer with $16 \times 16$ patch size, a deep learning architecture that leverages transformer models for image classification tasks. We use the Imagenet21k weights from the Huggingface endpoint google/vit-large-patch32-224-in21k as weight initialization.

- ViT-L.32 is a Vision Transformer with 32x32 patch size, a deep learning architecture that leverages transformer models for image classification tasks. We use the Imagenet21k weights from the Huggingface endpoint google/vit-large-patch32-224-in21k as weight initialization.

- ViT-G.14 is a Vision Transformer with 14x14 patch size, a deep learning architecture that leverages transformer models for image classification tasks. We use the Imagenet21k weights from the Huggingface endpoint google/vit-large-patch32-224-in21k as weight initialization.

The model and training hyperparameter setup is described in Table 5.

C.1.4. CONVERTING PRETRAINED MHA PARAMETERS TO APPROXIMATE ATTENTION PARAMETERS

**MHA to GQA**    To transform MHA $W_{Q,K,V,O}$ to their GQA counterparts, we observe that $W_{Q,O}$ do not have to be altered. For $W_{K,V}$, we reshape the matrices in tensor form ($n_{\mathrm{H}} \times d_{\mathrm{H}} \times d_{\mathrm{model}}$), and keep each $n_{\mathrm{KV}}$ KV heads, where $n_{\mathrm{KV}}$ is the number of KV heads of GQA. Note, that this approximation can be refined by selecting the $n_{\mathrm{KV}}$ heads that approximate $W_{K,V}$ best, e.g. with a discrete empirical interpolation method (DEIM) (Chaturantabut & Sorensen, 2010). We obtain MQA by setting $n_{\mathrm{KV}} = 1$.

**MHA to MLA**    We compute a truncated singular value decomposition (SVD) with rank $d_{\mathrm{c}}$ for $W_{Q,K,V}$ to obtain the down and up projections. For MLA with shared KV down projection, we discard the value down-projection and use the key downprojection as the shared one. Note that corresponding to the original deepseek implementation (DeepSeek-AI et al., 2024), and the deepseekV3 architecture, we do not compress $W_O$.

**MHA to Tucker Attention**    We compute $W_i := W_i^{\mathrm{Q}} W_i^{K,\top} \in \mathbb{R}^{d_{\mathrm{model}} \times d_{\mathrm{model}}}$ per head $i$. Analogously value and output matrices form a rank $d_v$ factorization of the *post-softmax weight* $\widetilde{W}_i := W_i^V W_i^O \in \mathbb{R}^{d_{\mathrm{model}} \times d_{\mathrm{model}}}$. Then we perform a high order singular value decomposition (HOSVD) of the resulting $\mathcal{W} = (W_i)_{i=1}^{n_{\mathrm{H}}} \in \mathbb{R}^{n_{\mathrm{H}} \times d_{\mathrm{model}} \times d_{\mathrm{model}}}$ and $\widetilde{\mathcal{W}} = (\widetilde{W}_i)_{i=1}^{n_{\mathrm{H}}} \in \mathbb{R}^{n_{\mathrm{H}} \times d_{\mathrm{model}} \times d_{\mathrm{model}}}$ with ranks $[r_1, r_2, r_3]$. Note that shared KV downprojection is possible by sharing $U_3$ between the Tucker factorizations of $\widetilde{\mathcal{W}}$ and $\mathcal{W}$.

## C.2. Training GPT2 from scratch

C.2.1. OPENWEBTEXT

We use the OpenWebText Corpus (Gokaslan & Cohen, 2019) to pretrain the GT2 variants. We tokenize the corpus using tiktoken v0.12.0 with gpt2 encoding in the data-preparation step. We use the same tokenizer for OpenWebText2.

C.2.2. TINYSHAKESPEARE

We use the TinyShakespeare dataset (Karpathy, 2015) for character-level language modeling experiments. This corpus consists of approximately 1MB of concatenated works of William Shakespeare, comprising roughly 1 million characters. We tokenize the corpus using tiktoken v0.12.0 with gpt2 encoding in the data-preparation step.

C.2.3. GPT2

Our implementation of GPT2 (Radford et al., 2019) is based on Andrey Karpathy's NanoGPT implementation, see `https://github.com/karpathy/nanoGPT`. The model and training hyperparameter setup is described in Table 5. We use context length $N = 1024$.

C.2.4. USAGE OF ROPE

The baseline version of GPT2 (called w/o RoPE in Table 2) uses the standard cosine position embedding in the token-dimension-to-model-dimension embedding layer. To evaluate the effects of RoPE on GPT2 with MhA,GQA, MLA, and

*Table 5.* Model and training hyperparameters

| Hyperparameter | ViT-B.16 | ViT-L.32 | ViT-G.14 | GPT2 | LLaMA3-1b |
|---|---|---|---|---|---|
| $d_{\mathrm{model}}$ | 768 | 1024 | 1664 | 768 | 2048 |
| $n_{\mathrm{H}}$ | 12 | 16 | 16 | 12 | 32 |
| $d_{\mathrm{H}}$ | 64 | 64 | 104 | 64 | 64 |
| $N$ | 196 | 49 | 256 | 1024 (256 for Shakespeare) | 4096 |
| Batch Size (Cifar10) | 128 | n.a. | n.a. | n.a. | n.a. |
| Batch Size (Cifar100) | 128 | n.a. | n.a. | n.a. | n.a. |
| Batch Size (ImageNet1k) | n.a | 512 | 512 | n.a. | n.a. |
| Batch Size (OpenWebText) | n.a. | n.a. | n.a. | 480 | n.a. |
| Batch Size (Shakespeare) | n.a. | n.a. | n.a. | 64 | n.a. |
| Batch Size (OpenWebText2) | n.a. | n.a. | n.a. | n.a. | 128 |
| Learning Rate | 1e-4 | 1e-3 | 1e-3 | 1e-4 | 3e-4 |
| Learning Rate Scheduler | | | Cosine Annealing w/ Warmup | | |
| Learning Rate Warmup | 5% | 5% | 5% | 0.5% | 1% |
| Number of Epochs (Cifar10) | 20 | n.a | n.a | n.a | n.a |
| Number of Epochs (Cifar100) | 20 | n.a | n.a | n.a | n.a |
| Number of Epochs (ImageNet1k) | n.a | 10 | 10 | n.a | n.a |
| Number of Iterations (OpenWebText) | n.a | n.a | n.a | 600000 | n.a |
| Number of Iterations (Shakespeare) | n.a | n.a | n.a | 10000 | n.a |
| Number of Iterations (OpenWebText2) | n.a | n.a | n.a | n.a | 100000 |
| L2 Regularization | 0.15 | 0.001 | 0.001 | 0.1 | 0.1 |
| Optimizer | AdamW | AdamW | AdamW | (fused) AdamW | AdamW |
| Parameters (baseline model) | 86M | 304M | 1.8B | 117M | 1.23B |

Tucker Attention, we remove the standard cosine position embedding and instead equip the attention variants with RoPE as follows, where $R(\ell, d)$ is the RoPE matrix for token position $\ell$ and dimension $d$.

**MHA**    We use RoPE as described in (Su et al., 2023), i.e.

$$X_m W_i^{\mathrm{Q}} R(m, d_{\mathrm{H}})(X_n W_i^{\mathrm{K}} R(n, d_{\mathrm{H}}))^{\top} \tag{16}$$

**GQA**    We use RoPE similar to the MHA case, i.e.

$$X_m W_i^{\mathrm{Q}} R(m, d_{\mathrm{H}})(X_n W_g^{\mathrm{K}} R(n, d_{\mathrm{H}}))^{\top} \tag{17}$$

the query RoPE matrix is evaluated for each query head, and the key RoPE matrix once for each head-group and subsequently broadcasted.

**Tucker**    We use RoPE as described in Equation (12), i.e.,

$$(\hat{Q}_m \times_3 R(m, r_3)^{\top}) \times_3 (K_n R(n, r_3)). \tag{18}$$

Note that the query and key RoPE matrix is evaluated once for all heads, respectively, which is more obvious from an equivalent formulation.

$$\hat{Q}_m \times_3 (K_n R(n, r_3) R(m, r_3)^{\top}). \tag{19}$$

**MLA - coupled RoPE**    Following the conclusion of Section 3.2.1, compute RoPE in the key latent dimension of MLA using, i.e.,

$$X_m W^{DQ} W_i^{UQ} W_i^{UV} R(m, d_{\mathrm{c}}) R^{\top}(n, d_{\mathrm{c}}) W^{DKV} X_n, \tag{20}$$

**MLA - decoupled RoPE**    We follow the canonical implementation of (DeepSeek-AI et al., 2024);

Instead of applying rotary position embeddings to all query/key dimensions, only a subset of channels undergoes rotational transformations, while the remaining channels remain purely semantic. For a sequence of $N$ tokens, the query and key matrices are partioned as

$$W^Q = [\, W^Q_{\mathrm{sem}} | W^Q_{\mathrm{rot}} \,] \in \mathbb{R}^{d_{\mathrm{model}} \times d_{\mathrm{model}}}, \quad \text{and} \quad W^K = [\, W^K_{\mathrm{sem}} | W^K_{\mathrm{rot}} \,] \in \mathbb{R}^{d_{\mathrm{model}} \times d_{\mathrm{model}}},$$

with $W^Q_{\mathrm{sem}}, W^K_{\mathrm{sem}} \in \mathbb{R}^{d_{\mathrm{model}} \times d_s}$, $W^Q_{\mathrm{rot}}, W^K_{\mathrm{rot}} \in \mathbb{R}^{d_{\mathrm{model}} \times d_r}$, where $d_s + d_r = d_{\mathrm{model}}$ denotes the concatenation. While $W^Q_{\mathrm{sem}}, W^K_{\mathrm{sem}}$ are further factorized as usual using MLA's low-rank factorization, RoPE is applied row-wise to the rotational components $W^Q_{\mathrm{rot}}, W^K_{\mathrm{rot}}$ just like in the MHA case,

$$A_{\mathrm{rot},i} = X_m W^Q_{\mathrm{rot},i} R(m, d_{\mathrm{H}})(X_n W^K_{\mathrm{rot},i} R(n, d_{\mathrm{H}}))^\top \tag{21}$$

Thus, the pre-softmax attention map for head $i$ consists of semantic and positional components as $A_i = Q_{\mathrm{sem},i} K^\top_{\mathrm{sem},i} + A_{\mathrm{rot},i}$.

### C.3. LLaMA-1B on OpenWebText2 training from scratch

**Setup**    We train a LLaMA 3-style decoder-only transformer from scratch on OpenWebText2 using GPT-NeoX. The model has $L = 16$ layers with hidden size $d_{\mathrm{model}} = 2048$, SwiGLU MLP width 8192 (rounded to a multiple of 256), and RMSNorm ($\varepsilon = 10^{-5}$). We use $n_{\mathrm{H}} = 32$ query heads with head dimension $d_{\mathrm{H}} = 64$ and grouped-query attention (GQA) with $n_{\mathrm{KV}} = 8$ key/value heads. Positional information is injected via RoPE with full rotary coverage (`rotary_pct`= 1.0), maximum context length $N = 4096$, and rotary base $\theta = 5 \cdot 10^5$. The Tucker Attentionimplementation uses the proposed latent RoPE. Following LLaMA conventions, we disable biases in attention, MLP, and normalization layers, and we tie input/output embeddings (`no_weight_tying`=false). Tokenization uses `cl100k_base`.

All runs use identical optimization and training hyperparameters. We train for $100,000$ iterations with Adam ($\beta_1 = 0.9, \beta_2 = 0.95, \epsilon = 10^{-8}$), peak learning rate $3 \cdot 10^{-4}$, minimum learning rate $3 \cdot 10^{-5}$, cosine decay over 10,000 iterations, and $1\%$ warmup. We apply weight decay 0.1 and gradient clipping at 1.0, with dropout disabled in both attention and MLP. Training is performed in BF16 precision. We use ZeRO stage 1 with partitioned optimizer states and overlapped communication, and enable activation checkpointing with per-layer checkpointing and partitioned activations. We use FlashAttention2 and keep other kernel fusions disabled for simplicity. Data is consumed via memory-mapped (`mmap`) shards. Validation is run every 500 iterations for 10 evaluation iterations. MLA and Tucker Attentionuse the latent RoPE positional encoding.

For the decoding timings, we use $N = 16k$ KV cache length, and query token lenght 1 to simulate autoregressive roll-out with exisiting KV cache. All timings are measured with the megatron timer of GPT-NeoX.

### C.4. Ablation study: GPT2 on Shakespeare

Table 6 we provide an ablation study for GPT2 trained from scratch on the Shakespeare dataset using the hyperparameters reported in Table 5.

# D. Additional Figures

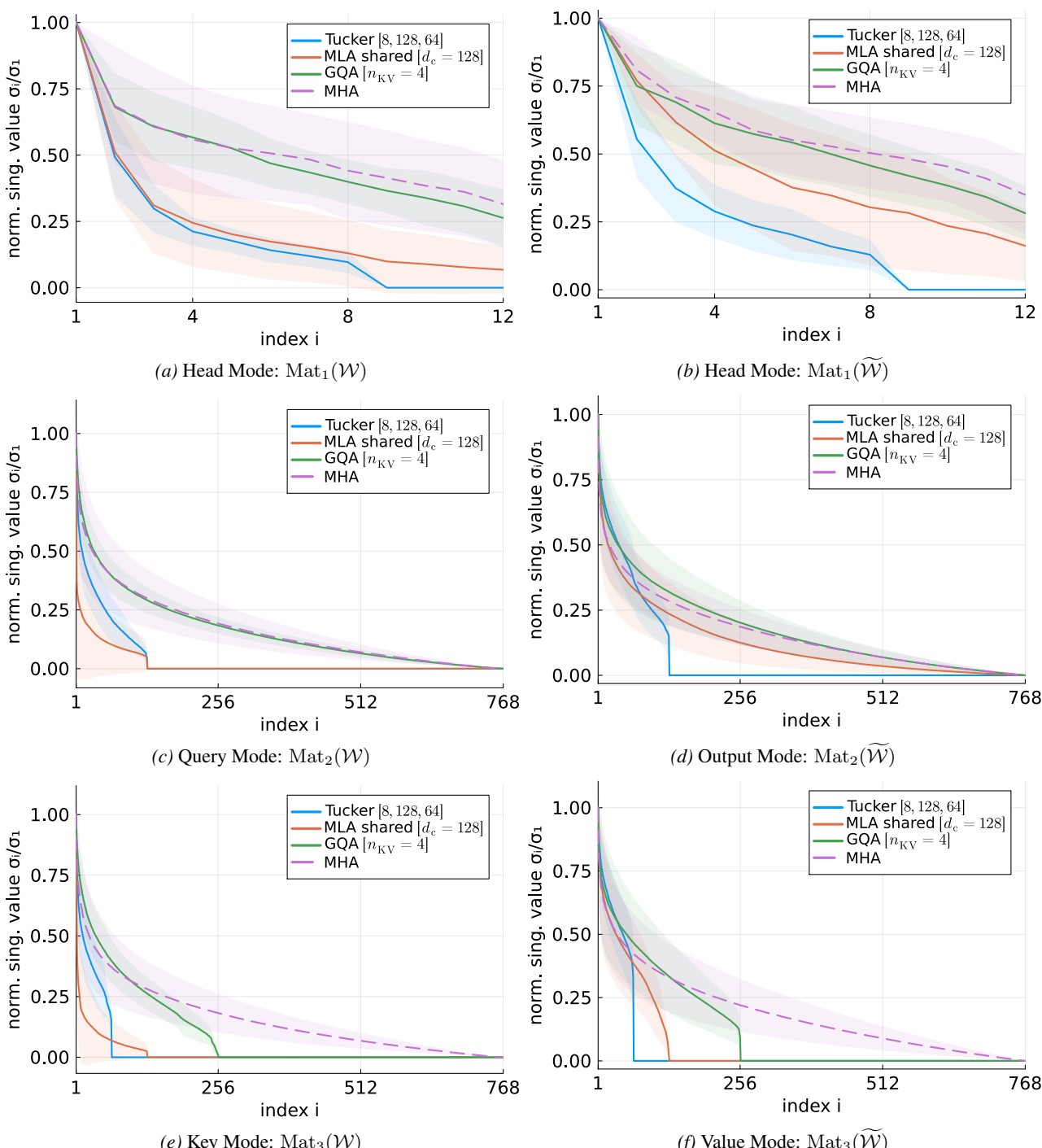

*Figure 5.* Normalized singular spectrum of the transformer layers in GPT2 after training. The singular spectrum was computed via a matricization of the tensor along a single mode; specific information about which mode is given in each subplot. The ribbon plots are calculated as a sample distribution over all twelve transformer layers in GPT2.

Table 6. Results for GPT-2 [$d_{model} = 768$, $n_H = 12$, $d_H = 64$] variants trained on Shakespeare with identical training hyperparameters. For each RoPE setting, we mark the best KV cache and parameter count to accuracy ration of GQA, MLA, and Tucker Attention bold. Tucker Attention achieves superior parameter efficiency ratios. MLA w/ Tucker-style RoPE refers to the method described in Equation (14).

| Method | Attn params [MB] | | KV Cache [MB] | | Val. loss | | Val. loss × Attn params | | Val. loss × KV Cache | |
|---|---|---|---|---|---|---|---|---|---|---|
| | w/o RoPE | w/ RoPE | w/o RoPE | w/ RoPE | w/o RoPE | w/ RoPE | w/o RoPE | w/ RoPE | w/o RoPE | w/ RoPE |
| MHA | 56.62 | | 9.000 | | 3.855 | 3.798 | 218.27 | 215.04 | 34.70 | 34.18 |
| GQA [$n_{KV} = 4$] | 37.74 | | 3.000 | | 3.859 | 3.798 | 145.64 | 143.34 | 11.58 | 11.39 |
| GQA [$n_{KV} = 2$] | 33.02 | | 1.500 | | 3.872 | 3.814 | 127.85 | 125.94 | 5.81 | 5.72 |
| GQA [$n_{KV} = 1$] | 30.66 | | 0.750 | | 3.875 | 3.809 | 118.81 | 116.78 | 2.91 | 2.86 |
| MLA separated K,V [$d_c^K = 128$] | 28.36 | 28.24 | 1.500 | 1.546 | 3.865 | 3.821 | 109.61 | 107.91 | 5.80 | 5.91 |
| MLA separated K,V [$d_c^K = 64$] | 21.28 | 21.24 | 0.750 | 0.796 | 3.884 | 3.834 | 82.65 | 81.43 | 2.91 | 3.05 |
| MLA separated K,V [$d_c^K = 32$] | 17.74 | 17.73 | 0.375 | 0.420 | 3.927 | 3.877 | 69.66 | 68.74 | 1.47 | 1.63 |
| MLA shared K,V [$d_c^K = 128$] | 25.96 | 25.86 | 0.750 | 0.773 | 3.875 | 3.823 | 100.59 | 98.86 | 2.91 | 2.96 |
| MLA shared K,V [$d_c^K = 64$] | 20.06 | 20.04 | 0.375 | 0.398 | 3.913 | 3.862 | 78.49 | 77.39 | 1.47 | 1.54 |
| MLA shared K,V [$d_c^K = 32$] | 17.10 | 17.14 | 0.187 | 0.210 | 3.974 | 3.967 | 67.96 | 67.99 | 0.74 | 0.83 |
| MLA shared K,V; latent RoPE [$d_c^K = 128$] | 25.96 | 26.00 | 0.750 | 0.750 | 3.875 | 3.905 | 100.59 | 101.53 | 2.91 | 2.93 |
| MLA shared K,V; latent RoPE [$d_c^K = 64$] | 20.06 | 20.10 | 0.375 | 0.375 | 3.913 | 3.952 | 78.49 | 79.44 | 1.47 | 1.48 |
| MLA shared K,V; latent RoPE [$d_c^K = 32$] | 17.10 | 17.16 | 0.187 | 0.187 | 3.974 | 4.025 | 67.96 | 69.07 | 0.74 | 0.75 |
| Tucker separated [8, 128, 128] | 15.74 | | 1.500 | | 3.877 | 3.814 | 61.02 | 60.03 | 5.82 | 5.72 |
| Tucker separated [8, 64, 64] | 6.30 | | 0.750 | | 3.908 | 3.849 | 24.62 | 24.25 | 2.93 | 2.89 |
| Tucker separated [6, 64, 64] | 5.92 | | 0.750 | | 3.929 | 3.864 | 23.26 | 22.87 | 2.95 | 2.90 |
| Tucker separated [6, 32, 32] | 2.66 | | 0.375 | | 3.987 | 3.919 | 10.61 | 10.42 | 1.50 | 1.47 |
| Tucker separated [4, 32, 32] | 2.56 | | 0.375 | | 3.989 | 3.934 | 10.21 | 10.07 | 1.50 | 1.48 |
| Tucker shared K,V [8, 128, 128] | 13.40 | | 0.750 | | 3.888 | 3.822 | 52.10 | 51.21 | 2.92 | 2.87 |
| Tucker shared K,V [8, 64, 64] | 5.14 | | 0.375 | | 3.951 | 3.881 | 20.31 | 19.95 | 1.48 | 1.46 |
| Tucker shared K,V [6, 64, 64] | 4.74 | | 0.375 | | 3.955 | 3.891 | 18.75 | 18.44 | 1.48 | 1.46 |
| Tucker shared K,V [6, 32, 32] | 2.08 | | 0.375 | | 4.058 | 3.974 | 8.44 | 8.27 | 1.52 | 1.49 |
| Tucker shared K,V [4, 32, 32] | 1.98 | | 0.187 | | 4.070 | 4.004 | 8.06 | 7.93 | 0.76 | 0.75 |

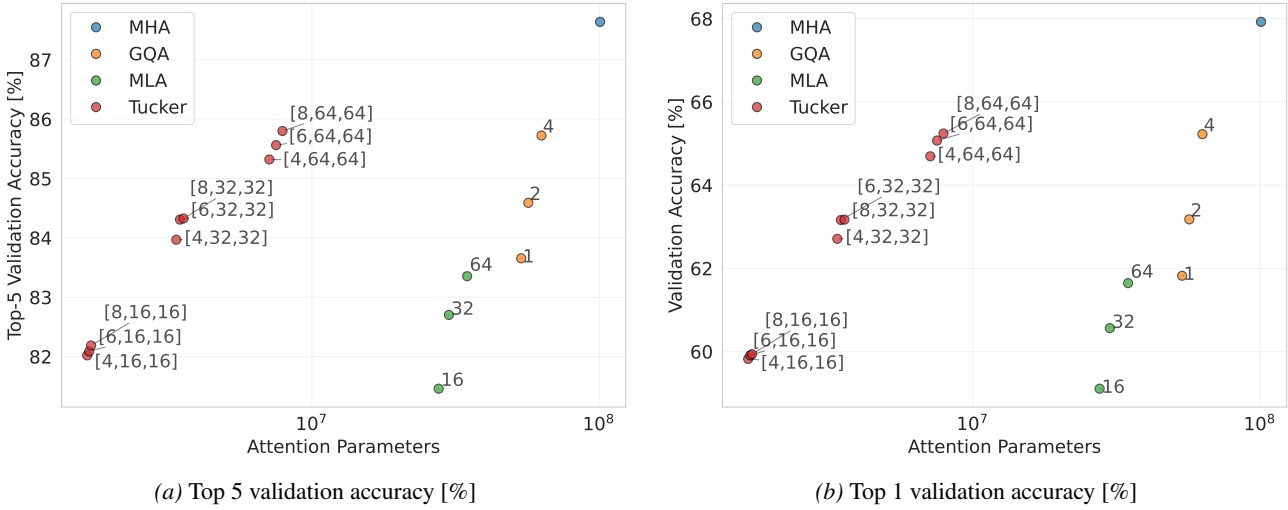

*(a)* Top 5 validation accuracy [%]

*(b)* Top 1 validation accuracy [%]

*Figure 6.* Vit-L 32 on Imagenet1k. Results denoting mean over 5 runs. The attention layer of ViT-L.32 has dimensions $d_{\mathrm{model}} = 1024, n_{\mathrm{H}} = 16, d_{\mathrm{H}} = 64$.

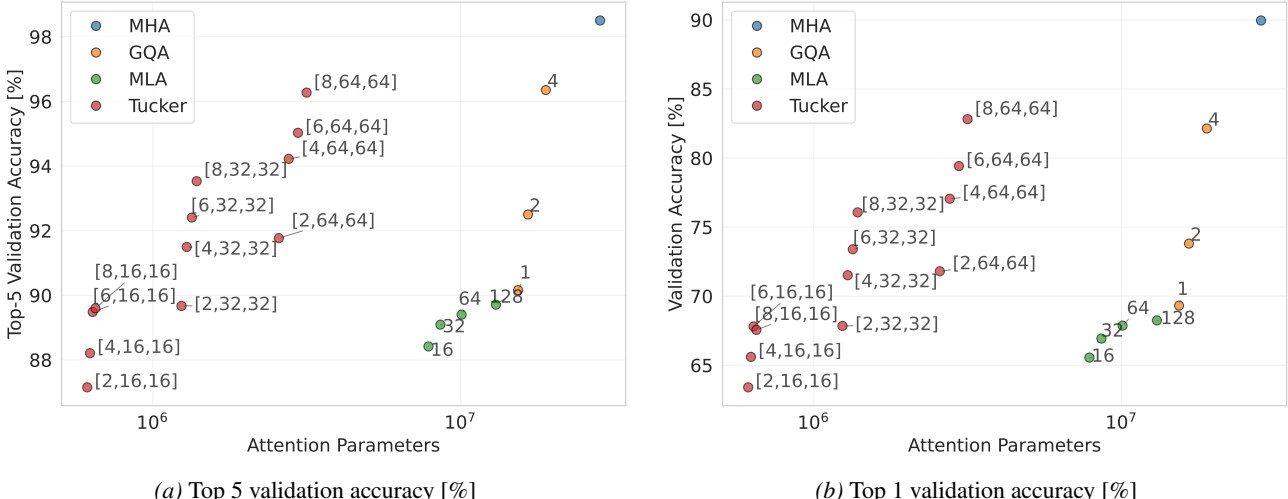

*(a)* Top 5 validation accuracy [%]

*(b)* Top 1 validation accuracy [%]

*Figure 7.* Vit-B 16 on Cifar100. Results denoting mean over 5 runs. The attention layer of Vit-B 16 has dimensions $d_{\mathrm{model}} = 768, n_{\mathrm{H}} = 12, d_{\mathrm{H}} = 64$.

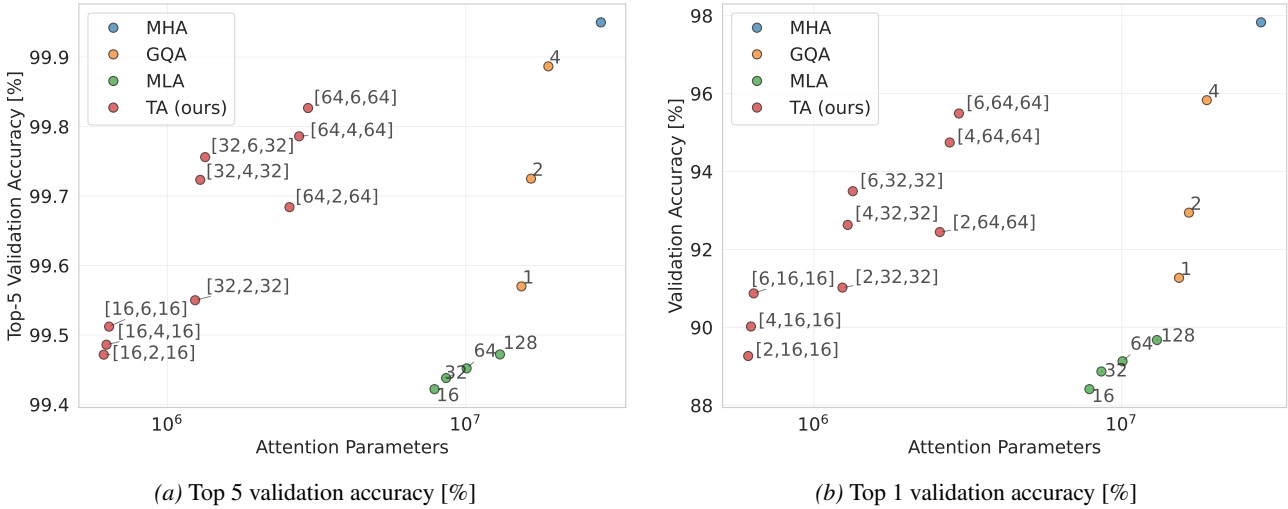

*(a)* Top 5 validation accuracy [%]          *(b)* Top 1 validation accuracy [%]

*Figure 8.* Vit-B 16 on Cifar10. Results denoting mean over 5 runs. The attention layer of Vit-B 16 has dimensions $d_{\mathrm{model}} = 768, n_{\mathrm{H}} = 12, d_{\mathrm{H}} = 64$.

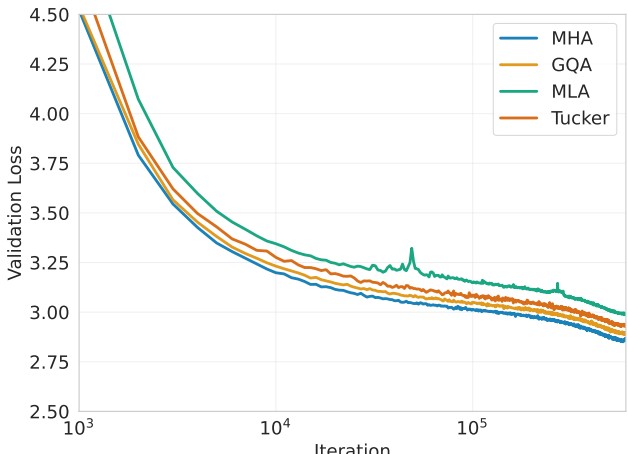

*Figure 9.* GPT2 validation cross-entropy loss on OpenWebtext without RoPE. Approximate attention methods are MLA with $d_c^{\mathrm{Q}} = d_c^{\mathrm{K}} = 128$, GQA with $n_{\mathrm{KV}} = 2$, Tucker Attention with $r = [8, 64, 128]$. Parameter counts are (in millions) MHA: 28.31, GQA: 16.51, MLA: 13.00, Tucker: 5.32. Tucker Attention converges well, despite having only a fraction of the trainable parameters of the other methods.

