# OpenReview forum: "Tucker Attention: A generalization of approximate attention mechanisms"
_ICML.cc/2026/Conference — ICML 2026 regular_

### Official Review · Reviewer_wRuK · 2026-02-16

**Soundness:** 3
**Presentation:** 2
**Significance:** 3
**Originality:** 3
**Overall Recommendation:** 4
**Confidence:** 3

**Summary:**

The paper change the multi-head attention as a tensor with softmax weight objects and proposes a low-rank decomposition as Tucker Attention that generalizes MHA, MQA, GQA, and MLA.

**Compliance With Llm Reviewing Policy:**

Affirmed.

**Key Questions For Authors:**

Is it possible to evaluate the proposed method on large-scale LLMs?

How does Tucker Attention interact and integrate with other low-ranking and adaptation methods like pruning, quantization, or LoRA?

**Limitations:**

Yes. The paper primarily proposes a structural modification to the attention mechanism. The mentioned limitations point to the existing transformer-based systems rather than show Tucker Attention's limitations itself.

**Strengths And Weaknesses:**

strengths:

Definition of a clean reformulation of multi-head attention as a Tucker Attention.

Parameters and memory reduction by the low-rank Tucker decomposition and maintaining strong performance.

weaknesses:

Figure 1 has not high quality.

Is Tucker Attention performs well and scalable in large-scale LLMs?

There aren't comprehensive analysis on rank selection.

---

> ### Author Rebuttal · Authors · 2026-03-30
>
> ### Response to Figure 1
>
> We have updated Figure 1 with a higher quality version
>
> ### Q on large-scale LLMs
>
> We have included an evaluation of the LLaMA3-1B LLM architecture (4k sequence length), implemented with GPT-NeoX. Using the megatron backend timers, we compare wall-time cost of the forward-pass, backward-pass, optimizer timings, and total iteration time in milliseconds during training on 2 nodes of 2 Nvidia H100 each, decoding is measured on 1 H100. The attention computation uses flash-attention for all parameter representations, the timings are measured end-to-end for the entire LLM, including non-compressed encoder and MLP layers.
>
> We observe: Tucker Attention uses only 10–20\% of the LLaMA3 baseline (GQA, $n_{\textrm{KV}}=8$)  attention parameters, with a modest 1–3\% increase in validation cross-entropy loss. It reduces the KV cache by 4–8$\times$, depending on rank. This reduction in parameters and KV cache lowers training cost, decreasing iteration wall time by up to 22\% vs. MHA and 17\% vs. the GQA (GQA, $n_{\textrm{KV}}=8$) baseline.
>
> Below is the table supporting our findings:
>
>
> | Method | Attn Params [MB] | PPL ↓ | Val loss ↓ |  Train Iter [ms] ↓ | KV Cache [MB] | Peak Decode Mem [GB] ↓ | Decode Latency [ms] ↓ |
> |---|---:|---:|---:|---:|---:|---:|---:|
> | MHA | 268 | 12.87 | 2.55 |     5.19 | 268 | 6.5 | 1.379 |
> | GQA $n_{KV}=8$ | 168 | 13.62 | 2.61 |    4.87 | 67 | 5.6 | 1.310 |
> | GQA $n_{KV}=2$ | 142 | 13.87 | 2.63 |     4.73 | 16.7 | 5.6 | 1.305 |
> | MQA $n_{KV}=1$ | 138 | 14.27 | 2.64 |     4.67 | 8.38 | 5.6 | 1.296 |
> | MLA $d_c^K,d_c^Q=128$ | 93.3 | 14.54 | 2.68 |  4.42      | 16.7 |  5.9|1.341|
> | MLA $d_c^K,d_c^Q=64$ | 79.2 | 14.77 | 2.69 |     4.36    | 8.38 |  5.6|1.317 |
> | MLA $d_c^K,d_c^Q=32$ | 73.4 | 15.65 | 2.75 |  4.30   | 4.19 |  5.4 | 1.292|
> | Tucker $32,128,128$ | 33.5 | 14.29 | 2.65 | 4.41 | 16.7 | 5.4 | 1.306 |
> | Tucker $32,128,64$ | 21.0 | 14.60 | 2.68 | 4.33 | 8.38 | 5.2 | 1.294 |
> | Tucker $32,64,64$ | 12.6 | 14.76 | 2.69 | 4.05 | 8.38 | 5.2 | 1.274 |
>
>
>
>
> We have further extended the ViT experiments to ViT14g at 1.8B parameters. We observe the same effect as in the smaller ViT models, i.e. the Tucker attention achieves higher parameter efficiency than baseline methods, see table below:
>
> | Method | Attention Parameters ↓ | Top-5 Accuracy ↑ |
> |---|---:|---:|
> | MHA | 210M | 0.9207 |
> | GQA (n_kv=4) | 132M | 0.9001 |
> | GQA (n_kv=2) | 118M | 0.8952 |
> | GQA (n_kv=1) | 111M | 0.8871 |
> | MLA (d_c=64) | 66.8M | 0.8773 |
> | MLA (d_c=32) | 59.3M | 0.8679 |
> | MLA (d_c=16) | 56.4M | 0.8519 |
> | Tucker (4-64-64) | 11.4M | 0.9001 |
> | Tucker (6-64-64) | 12.1M | 0.9047 |
> | Tucker (8-64-64) | 12.6M | 0.9052 |
> | Tucker (4-32-32) | 5.55M | 0.8911 |
> | Tucker (6-32-32) | 5.68M | 0.8941 |
> | Tucker (8-32-32) | 5.81M | 0.8963 |
> | Tucker (4-16-16) | 2.73M | 0.8722 |
> | Tucker (6-16-16) | 2.76M | 0.8691 |
> | Tucker (8-16-16) | 2.80M | 0.8754 |
>
>
>
> ### Response to analysis of rank selection
>
> We underline Table 5 in the Appendix which performs an ablation study of Tucker Attention~over a range of ranks. While we agree that rank selection, which adds an additional set of hyperparameters, is a non-trivial task, we comment that the rank analysis present in Section 2.1 can guide rank selection based on domain experience with other attention mechanisms. Specifically, selection of $r_3$ (key mode) can be initialized through a GQA lens, i.e, $r_3=d_H n_KV$ or an MLA lens, i.e., $r_3=d_c^K$
>
> ### Q on pruning, quantization, and LoRA
>
> On pruning and quantization:  Since our tensor approach does not alter the overall attention mechanism, conventional pruning and quantization approaches can be applied to Tucker Attention without modification.
>
> Tucker Attention modifies the pre- and post-attention weight parameterization, so LoRA adapters should be applied to the Tucker factors in Eq.~(7) rather than to the usual query, key, value, and output matrices. In practice, one can adapt (a) only the core $C$, keeping the pre-trained Tucker bases fixed while updating the coefficients, or (b) both the core and bases $U$, allowing changes to both the subspace and its coefficients. This yields an interpretable LoRA integration for Tucker Attention.
>
> ### Response to limitation
>
> Tucker Attention shares common limitations with other attention mechanisms, especially the need to select appropriate rank dimensions as hyperparameters. This challenge is not unique to our method, and our ability to recover widely used attention variants, where extensive hyperparameter tuning is already available, helps guide rank selection. Tucker Attention can also be substantially sped up with a dedicated GPU kernel that exploits its reduced parameter count; even without such optimization, it remains compatible with and can outperform approaches that use optimized kernels (see new Llama results). We are happy to add this discussion to the paper.

---

> > ### Author Rebuttal · Reviewer_wRuK · 2026-04-02
> >
> > The authors acknowledge the limitations cited in the review. My score remains unchanged

---

> > > ### Author Response · Authors · 2026-04-05
> > >
> > > We thank the reviewer for their response and fully respect the reviewer's decision to not raise their score. As reason, the reviewer states "The authors acknowledge the limitations cited in the review." Unfortunately, we do not fully understand this statement and wish to better understand the reviewer's comment to be able to improve our paper. The weaknesses mentioned in the review were the quality of Figure 1 which is easily fixed (please note that we cannot upload a new version of our manuscript at this point), the size of the numerical experiments (we now provide results for large scale architectures as discussed in length in our response and that are easily integrated into the paper), and a missing analysis for rank selection (which we provided in the original manuscript as pointed out). We do acknowledge the initially missing numerical experiments on large scale architectures and that the image quality of Figure 1 can be improved, but we feel like these weaknesses have been resolved and can easily be integrated in our manuscript.

---

### Official Review · Reviewer_e5S7 · 2026-03-04

**Soundness:** 4
**Presentation:** 4
**Significance:** 4
**Originality:** 4
**Overall Recommendation:** 6
**Confidence:** 3

**Summary:**

The task the article considers is parameter-efficient attention computation in Transformer architectures. The analysis of the authors is based on the observation that multi-head attention (MHA) is a specific factorization of the pre- and post-softmax weight tensors. Whereas the earlier approximate attention computation methods, such as MLA, MQA, and GQA are low-rank approximations of the MHA, the key idea of the article is that the pre- and post-softmax tensors can be factorized directly by a Tucker decomposition. The authors also show that MLA, MQA, and GQA are special cases of the proposed method and, unlike them, the proposed decomposition  enables also compressing the head mode of the attention tensor.

The experiments show that the proposed method is significantly more parameter-efficient than the earlier low-rank approximate attention computation methods, and according to the authors, it is orthogonal and can be combined with the other types of approximate attention mechanisms, such as linear attention and flash attention.

**Compliance With Llm Reviewing Policy:**

Affirmed.

**Final Justification:**

The article presents a unified view of earlier low-rank approximate attention mechanisms, and proposes a powerful generalization based on this unified framework. In particular, this enables compression over the head mode of the attention tensors unlike the earlier methods. The paper was an enjoyable read, and the idea is clever. As I did not have any significant concerns about the paper, I maintain my positive score.

**Key Questions For Authors:**

I do not have any questions for the authors.

**Limitations:**

Yes.

**Strengths And Weaknesses:**

- The analysis of the article is valid as far as I can tell, and the experiments support the main claim about the parameter efficiency of the proposed method.

- The paper is well-written, and the notation is consistent.

- Parameter-efficient attention computation is a research problem with a high impact because of the wide-spread use and high-computational cost of Transformer architectures, and the proposed method is significantly more parameter efficient than the earlier low-rank approximate attention mechanisms. However, I am not hundred-percent sure how significant the savings are in practice if the other types of the approximate attention mechanisms, such as linear attention (that the authors claim are orthogonal with the proposed method) are also used.

- The key idea of directly factorizing the pre- and post-softmax weight tensors instead of approximating MHA is ingenious. The article identifies earlier low-rank approximate attention computation mechanisms as special cases of a more general formulation, and shows that this more general formulation yields a more efficient algorithm. In summary the analysis of the article is highly original, since the article both provides valuable insights into the earlier methods, and delivers a more powerful generalization.

---

> ### Author Rebuttal · Authors · 2026-03-30
>
> ### Response to linear attention
>
> We sincerely thank the reviewer for their encouraging feedback. To briefly reply to your remark on linear attention, we note that many state-of-the-art LLMs adopt GQA or MLA variants. Models incorporating linear attention mechanisms (e.g., Mamba in Nemotron3) typically restrict these to a subset of layers. This work proposes a parameter-efficient and interpretable alternative to "quadratic" attention mechanisms as GQA and MLA. As a result, the approach is broadly applicable across attention layers in most contemporary LLMs, and remains compatible with models that mix attention types

---

> > ### Author Rebuttal · Reviewer_e5S7 · 2026-04-02
> >
> > Thank you for your response. I will maintain my positive score.

---

### Official Review · Reviewer_xHEb · 2026-03-10

**Soundness:** 2
**Presentation:** 2
**Significance:** 2
**Originality:** 3
**Overall Recommendation:** 3
**Confidence:** 3

**Summary:**

This paper proposes Tucker Attention, a Tucker-factorized formulation of attention that unifies several existing mechanisms, including MHA, MQA, GQA, and MLA. The main idea is to represent attention weights in a tensorized form and use Tucker factorization to obtain a more general parameterization that can also compress additional modes beyond those used in prior methods. The paper further discusses compatibility with practical components such as KV cache, RoPE, and FlashAttention, and presents experiments on GPT-2-scale models to evaluate the proposed approach.

**Compliance With Llm Reviewing Policy:**

Affirmed.

**Final Justification:**

The rebuttal addressed some of my concerns by adding more experimental evidence. However, my main concern still remains: the response mainly restates the novelty in subjective terms, rather than clearly establishing a substantial distinction from closely related tensorized attention and unifying reformulation works. In addition, the authors acknowledged that Figure 1 contained an error, which further weakens my confidence in the presentation. I decided to maintain my original score and increase my confidence score.

**Key Questions For Authors:**

See weaknesses.

**Limitations:**

See weaknesses.

**Strengths And Weaknesses:**

**Strengths:**

1. The paper provides a unifying framework of MHA, MQA, GQA, and MLA through Tucker factorization. I found it is conceptually useful and easy to follow.
2. The paper also discusses practical issues such as KV cache, RoPE, and FlashAttention compatibility, which improve the work more relevant from an implementation perspective.

**Weaknesses:**

**Major:**

1. The main contribution appears to be a unifying reformulation of existing approaches such as MLA and GQA. From this perspective, the novelty may be somewhat limited, and the paper reads more as an insightful generalization than as a clearly new attention mechanism with clear advantages across setting.
2. The empirical validation is limited for the scope of the claims, as the experiment are only performed on GPT-2 scale model.

**Minor:**

1. Typo in lines 62-63. "This proposed Tucker Attention has two key features;" It seems that a colon may be more appropriate.
2. What is the difference between Figure 1(a) and Figure 1(b)? As currently presented, the two subfigures look almost the same.

---

> ### Author Rebuttal · Authors · 2026-03-30
>
> ### Response to weakness on novelty.
>
> Tucker Attention provides a clearly new attention mechanism, and we see its ability to, for special choices of ranks, generalize to popular existing methods as a clear strength, not a weakness.  Showing the main advantages of Tucker Attention, i.e., formulating a low-rank tensor approach that for specially chosen ranks, will generalize to existing methods while at the same time allowing for compression in the head dimension, is not trivial and requires deep understanding and intuition about the underlying mathematical structure.
> As illustrated in Figure 1 and discussed in detail in Section 2.1, while Tucker Attention does generalize to popular techniques, its overall design differs substantially from existing approaches. In fact, its equivalence to existing approaches for careful rank choices is not trivial and has been discussed in Section 2.2, with the main derivations requiring deeper structural insights, see section B1 in the Appendix. Besides its compatibility with KV Caching, RoPE, and Flash Attention, the new abilities of Tucker attention, in particular its ability to efficiently compress the head and output modes (a property that none of the popular approaches share), leads to significant gains in parameter efficiency. These advantages have been presented clearly across settings.  Our vision transformer results (Section 4.1) show that Tucker Attention requires almost an order of magnitude fewer trainable parameters to achieve similar performances compared to GQA and MLA. The GPT2 on OpenWebText results of Section 4.2 show that Tucker Attention converges at the same rate as the canonical methods and reaches a lower loss than MLA while showing improved stability and a reduced parameter count. The results are validated on a smaller problem with an extensive hyperparameter study in Section 4.3 and on a larger 1B LLaMA 3 model (see next response).
>
> ###  Response to limited scale
>
> We have included an evaluation of the LLaMA3-1B LLM architecture (4k sequence lenght), implemented with GPT-NeoX. Using the megatron backend timers, we compare wall-time cost of the forward-pass, backward-pass, optimizer timings, and total iteration time in milliseconds during training on 2 nodes of 2 Nvidia H100 each, decoding is measured on 1H100. The attention computation uses flash-attention for all parameter representations, the timings are measured end-to-end for the entire LLM, including non-compressed encoder and MLP layers.
>
> We observe: Tucker Attention uses only 10–20\% of the LLaMA3 baseline  (GQA, $n_{\textrm{KV}}=8$)  attention parameters, with a modest 1–3\% increase in validation cross-entropy loss. It reduces the KV cache by 4–8$\times$, depending on rank. This reduction in parameters and KV cache lowers training cost, decreasing iteration wall time by up to 22\% vs. MHA and 17\% vs. the GQA (GQA, $n_{\textrm{KV}}=8$) baseline.
>
> Below is the table supporting our findings:
> | Method | Attn Params [MB] | PPL ↓ | Val loss ↓ |  Train Iter [ms] ↓ | KV Cache [MB] | Peak Decode Mem [GB] ↓ | Decode Latency [ms] ↓ |
> |---|---:|---:|---:|---:|---:|---:|---:|
> | MHA | 268 | 12.87 | 2.55 |     5.19 | 268 | 6.5 | 1.379 |
> | GQA $n_{KV}=8$ | 168 | 13.62 | 2.61 |    4.87 | 67 | 5.6 | 1.310 |
> | GQA $n_{KV}=2$ | 142 | 13.87 | 2.63 |     4.73 | 16.7 | 5.6 | 1.305 |
> | MQA $n_{KV}=1$ | 138 | 14.27 | 2.64 |     4.67 | 8.38 | 5.6 | 1.296 |
> | MLA $d_c^K,d_c^Q=128$ | 93.3 | 14.54 | 2.68 |  4.42      | 16.7 |  5.9|1.341|
> | MLA $d_c^K,d_c^Q=64$ | 79.2 | 14.77 | 2.69 |     4.36    | 8.38 |  5.6|1.317 |
> | MLA $d_c^K,d_c^Q=32$ | 73.4 | 15.65 | 2.75 |  4.30   | 4.19 |  5.4 | 1.292|
> | Tucker $32,128,128$ | 33.5 | 14.29 | 2.65 | 4.41 | 16.7 | 5.4 | 1.306 |
> | Tucker $32,128,64$ | 21.0 | 14.60 | 2.68 | 4.33 | 8.38 | 5.2 | 1.294 |
> | Tucker $32,64,64$ | 12.6 | 14.76 | 2.69 | 4.05 | 8.38 | 5.2 | 1.274 |
>
>
>
>
> We have further extended the ViT experiments to ViT14g at 1.8B parameters - we observe the same effect as in the smaller ViT models, i.e. the Tucker attention achieves higher parameter efficiency than baseline methods, see table below:
>
> | Method | Attention Parameters ↓ | Top-5 Accuracy ↑ |
> |---|---:|---:|
> | MHA | 210M | 0.9207 |
> | GQA (n_kv=4) | 132M | 0.9001 |
> | GQA (n_kv=2) | 118M | 0.8952 |
> | GQA (n_kv=1) | 111M | 0.8871 |
> | MLA (d_c=64) | 66.8M | 0.8773 |
> | MLA (d_c=32) | 59.3M | 0.8679 |
> | MLA (d_c=16) | 56.4M | 0.8519 |
> | Tucker (4-64-64) | 11.4M | 0.9001 |
> | Tucker (6-64-64) | 12.1M | 0.9047 |
> | Tucker (8-64-64) | 12.6M | 0.9052 |
> | Tucker (4-32-32) | 5.55M | 0.8911 |
> | Tucker (6-32-32) | 5.68M | 0.8941 |
> | Tucker (8-32-32) | 5.81M | 0.8963 |
> | Tucker (4-16-16) | 2.73M | 0.8722 |
> | Tucker (6-16-16) | 2.76M | 0.8691 |
> | Tucker (8-16-16) | 2.80M | 0.8754 |
>
> ### Response to typo on lines 62-63
>
> Fixed
>
> ### Response to Figure 1a
>
> We errantly linked the MHA figure (Figure 1a) with the GQA figure (Figure 1b).  This has now been fixed.

---

> > ### Author Rebuttal · Reviewer_xHEb · 2026-04-03
> >
> > While the authors have addressed some of my concerns by providing additional experimental results, I still find the overall presentation and novelty insufficient. I will maintain my original score.

---

> > > ### Author Response · Authors · 2026-04-05
> > >
> > > We thank the reviewer for their feedback and fully respect the reviewer's decision to not raise their score. As we understand the reviewer, this decision was reached since our work appears to be a unifying reformulation of existing approaches. We wish to again underline that our method generalizes to popular attention mechanisms (MHA, MLA, GQA, MQA) for carefully selected choices of the Tucker rank, which we see as one of the main strengths. It is not a reformulation of existing methods. Our methodology and its mathematical formulation is fundamentally different to these approaches as can directly be seen from, e.g., Figure 1. This also becomes clear by the fact that compared to these approaches, our method can compress the head dimension. This results in our method's improved performance across the large range of benchmarks with parameter compressions that the popular approaches cannot achieve.

---

### Official Review · Reviewer_vB7z · 2026-03-12

**Soundness:** 3
**Presentation:** 3
**Significance:** 3
**Originality:** 2
**Overall Recommendation:** 4
**Confidence:** 2

**Summary:**

The paper revisits the standard formulation of self-attention. Instead of viewing attention as four independent projection matrices $W_Q, W_K, W_v, W_O$. More precisely,  the authors reinterpret it as two structured third-order tensors: a pre-softmax tensor, capturing query–key interactions, and a post-softmax tensor, for values-to-outputs interactions. Building on this reformulation, the paper introduces Tucker Attention, which applies a Tucker decomposition directly to these tensors. With this perspective, several existing attention mechanisms—such as MHA, MQA/GQA, and MLA—can be recovered as particular cases corresponding to specific choices of tensor ranks. The main idea is that this tensor viewpoint reveals compressible structure that is not fully exploited by existing approximations, in particular along the head and output modes. As a consequence, the method has the nice proprty of reducing the number of attention parameters and the KV cache footprint while maintaining comparable accuracy.

 The authors also discuss the compatibility of the approach with common practical components such as KV caching, RoPE positional encoding, and FlashAttention implementations.
Empirically, experiments on Vision Transformers and GPT-2 style models suggest that Tucker Attention can achieve a favorable parameter–accuracy tradeoff compared to GQA and MLA.

**Compliance With Llm Reviewing Policy:**

Affirmed.

**Final Justification:**

My main concerns were adequately addressed during the rebuttal; however, I still consider the level of novelty to be moderate. For this reason, I have maintained my initial (slightly) positive score.

**Key Questions For Authors:**

Please refer to the Weaknesses Section.
* In the Experiemtnal Section, could you also report metrics around end-to-end system efficiency: wall-clock speed, GPU memory footprint during training, or end-to-end inference latency on real hardware?
* In the Experiemtnal Section, could you test Tucker Attention in larger-scale transformer models (e.g., >1B parameters)?
* Could you discuss more about the sensitivity to the multiple rank hyperparameters and give guidelines on how to choose them (impact on in the efficiency/memory footprint/accuracy trade-off,on potential optimisation instability, on potential failure modes)?

**Limitations:**

Yes, the authors adequately discussed the limitations and potential negative societal impact of their work

**Strengths And Weaknesses:**

**Strengths**

The main strength of the paper is to conceptually reframe the attention as a tensor object rather than independent projection matrices. This abstraction allows the authors to emphasize	how different attention variants compress different modes of the attention tensor and	how rank choices correspond to architectural design decisions. The paper does more than proposing another efficient attention block; it gives a unifying explanation of how MHA, MQA/GQA, and MLA sit inside a common tensor-factorization family.

A second strength is that the empirical results, within the paper’s scale, are fairly coherent. On ViT and GPT-2, Tucker variants often achieve similar validation behavior with materially fewer attention parameters;

A third strength is ability to extend the approach to take into deployment issues by considering KV cache, RoPE, and FlashAttention compatibility; in the same vein, the link between tensor ranks and memory footprint is stated clearly.

**Novelty**

The novelty is moderate rather than strong due to overlap with recent work on tensorized attention and tensor decompositions in transformers.
However, the paper would benefit from a clearer comparison with closely related approaches.(e.g. HEAT, TensorLLM, Tensor Product Attention, and TransMLA). For instance,  TensorLLM, published in 2025, also compresses multi-head attention through tensorization plus Tucker decomposition across heads, while Tensor Product Attention (also 2025) proposes a tensor-factorized attention mechanism that unifies several efficient-attention variants, is RoPE-compatible, and improves KV-cache efficiency.

**Weaknesses**

The paper mostly reports parameter counts (including for KV cache formulas), and accuracy/perplexity metrics. I would he been much more interseting to report metrics around end-to-end system efficiency: wall-clock speed,  GPU memory footprint during training, or end-to-end inference latency on real hardware.

The second weakness is scale. The experimental evaluation is restricted to GPT-2 LLMs and ViT benchmarks, and it does not seem to be enough to establish robustness for modern LLM regimes where GQA and MLA matter most. In other words, it is unclear whether the observed benefits would still persist at B-scale model training.

The third weakness is an insufficient investigation of the sensitivity to the multiple rank hyperparameters. How to choose the particular ranks remain unclear, especially in the efficiency/memory footprint/accuracy trade-off, as well as for training stability.

---

> ### Author Rebuttal · Authors · 2026-03-30
>
> ### Q on novelty
>
> We thank the reviewer for pointing out the related literature. We already cite most of these works and will make method names (HEAT, TensorLLM, Tensor Product Attention, TransMLA) more explicit in the final version.
>
> **Formulation.**
> Tucker Attention differs fundamentally in how attention weights are represented: HEAT factorizes individual weight matrices post hoc, TensorLLM stacks per-head weights into tensors but still operates at the full-rank head level, Tensor Product Attention (TPA) relies on a data-dependent factorization, and TransMLA reformulates specific attention layers (GQA→MLA) without a unified tensor view.
>
> **Generalization.**
> Tucker Attention provides a unified framework that recovers MHA, GQA, MQA, and MLA as special cases. HEAT does not address architectural generalization, TensorLLM does not generalize across attention variants, TPA only recovers MHA/GQA under constrained (non-contextual) settings, and TransMLA focuses on mapping GQA to MLA rather than unifying all variants.
>
> **Compression across heads.**
> Tucker Attention explicitly compresses along the head dimension via Tucker ranks. HEAT does not perform structured head-wise compression, TensorLLM factorizes within each head without compressing across heads, TPA uses CP-style decompositions without explicit multi-mode rank control over heads, and TransMLA does not introduce a general compression mechanism across heads.
>
> **Expressivity and analysis.**
> The joint Tucker structure provides a more expressive low-rank parameterization and enables mode-wise spectral analysis. In contrast, HEAT, TensorLLM, and TPA do not provide comparable multi-mode spectral insights, and TransMLA does not include such analysis or interpretability tools.
>
>
> ### Q on pamameter counts, report metrics, and scale
>
> We have included an evaluation of the LLaMA3-1B LLM architecture (4k sequence lenght), implemented with GPT-NeoX. Using the megatron backend timers, we compare wall-time cost of the forward-pass, backward-pass, optimizer timings, and total iteration time in milliseconds during training on 2 nodes of 2 Nvidia H100 each, decoding is measured on 1H100. The attention computation uses flash-attention for all parameter representations, the timings are measured end-to-end for the entire LLM, including non-compressed encoder and MLP layers.
>
> We observe: Tucker Attention uses only 10–20\% of the LLaMA3 baseline (GQA, $n_{\textrm{KV}}=8$) attention parameters, with a modest 1–3\% increase in validation cross-entropy loss. It reduces the KV cache by 4–8$\times$, depending on rank. This reduction in parameters and KV cache lowers training cost, decreasing iteration wall time by up to 22\% vs. MHA and 17\% vs. the GQA (GQA, $n_{\textrm{KV}}=8$) baseline.
>
> Below is the table supporting our findings:
>
> |Method|Attn Params [MB]|PPL ↓|Val loss ↓|Train Iter [ms] ↓|KV Cache [MB]|Peak Decode Mem [GB] ↓|Decode Latency [ms] ↓|
> |---|---:|---:|---:|---:|---:|---:|---:|
> |MHA|268|12.87|2.55|5.19|268|6.5|1.379|
> |GQA $n_{KV}=8$|168|13.62|2.61|4.87|67|5.6|1.310|
> |GQA $n_{KV}=2$|142|13.87|2.63|4.73|16.7|5.6|1.305|
> |MQA $n_{KV}=1$|138|14.27|2.64|4.67|8.38|5.6|1.296|
> |MLA $d_c^K,d_c^Q=128$|93.3|14.54|2.68|4.42|16.7|5.9|1.341|
> |MLA $d_c^K,d_c^Q=64$|79.2|14.77|2.69|4.36|8.38|5.6|1.317|
> |MLA $d_c^K,d_c^Q=32$|73.4|15.65|2.75|4.30|4.19|5.4|1.292|
> |Tucker $32,128,128$|33.5|14.29|2.65|4.41|16.7|5.4|1.306|
> |Tucker $32,128,64$|21.0|14.60|2.68|4.33|8.38|5.2|1.294|
> |Tucker $32,64,64$|12.6|14.76|2.69|4.05|8.38|5.2|1.274|
>
>
> We have further extended the ViT experiments to ViT14g at 1.8B parameters. We observe the same effect as in the smaller ViT models, i.e. the Tucker attention achieves higher parameter efficiency than baseline methods, see table below:
>
> |Method|Attention Parameters↓|Top-5Accuracy↑|
> |---|---:|---:|
> |MHA|210M|0.9207|
> |GQA(n_kv=4)|132M|0.9001|
> |GQA(n_kv=2)|118M|0.8952|
> |GQA(n_kv=1)|111M|0.8871|
> |MLA(d_c=64)|66.8M|0.8773|
> |MLA(d_c=32)|59.3M|0.8679|
> |MLA(d_c=16)|56.4M|0.8519|
> |Tucker(4-64-64)|11.4M|0.9001|
> |Tucker(6-64-64)|12.1M|0.9047|
> |Tucker(8-64-64)|12.6M|0.9052|
> |Tucker(4-32-32)|5.55M|0.8911|
> |Tucker(6-32-32)|5.68M|0.8941|
> |Tucker(8-32-32)|5.81M|0.8963|
> |Tucker(4-16-16)|2.73M|0.8722|
> |Tucker(6-16-16)|2.76M|0.8691|
> |Tucker(8-16-16)|2.80M|0.8754|
>
>
>
> ### Q on rank sensitivity and training stability
>
> We underline Figure 3 and Table 5 which give an empirical tradeoff between rank(parameter count) and accuracy/validation loss.  While rank selection is a non-trivial task, the rank analysis presented in Section 2.1 can guide rank selection based on domain experience with other attention mechanisms (e.g., MLA and GQA).
>
> We underline Figure 8 in the paper which shows the validation loss for GPT2 training.  The figure shows that compared to MLA, Tucker Attention has fewer spikes.  Moreover, we studied the training loss in LLaMA3 and found that Tucker Attention was overall more stable (fewer spikes) than GQA and MHA.

---

> > ### Author Rebuttal · Reviewer_vB7z · 2026-04-03
> >
> > Thank you for the detailed clarifications and additional results. My main concerns appear to be adequately addressed but I still consider the novelty as moderate. I will keep my (slightly) positive score.

---

> > > ### Author Response · Authors · 2026-04-05
> > >
> > > We thank the reviewer for their response and fully respect their decision to not raise their score. While the reviewer's concerns have been fully resolved, the score remains (slightly) positive as the novelty is considered moderate. As we understand the reviewer, this concerns novelties compared to HEAT, TensorLLM, Tensor Product Attention, and TransMLA that (except for HEAT) we all referenced in our original submission. While a large number of works have addressed parameter efficiency of attention formulations, the mentioned works are very different to our approach as we have detailed in our response. We are very happy to add a more detailed discussion to our manuscript.

---

### Decision · Program_Chairs · 2026-04-30

**Decision:**

Accept (regular)

**Comment:**

This paper presents Tucker Attention, a generalization of Multi Head Attention, Multi Head Latent attention and Grouped Query Attention. The idea is to represent the pre-softmax output of Multi Head Attention as the  tensor product of a tensor $\mathcal{W}\in n_H\times d\times d$  (here $n_H$ is the number of heads and $d$ is the token dimension) with the input sequence $X$ across the last two tensor dimensions. Under this formulation, the ordinary Multi Head Attention can be interpreted as a requirement that all the slices $\mathcal{W}_{i,\cdot,\cdot}$ be low rank. In contrast, the authors propose a general low Tucker rank structure. MHL, GQA and MLA are all particular cases of the method depending on the Tucker ranks in each dimension.

The authors demonstrate that the proposed method is more parameter efficient than existing methods whilst achieving comparable accuracy. Interestingly, they also demonstrate that existing attention modules exhibit spectral decay in the attention direction, though unlike the proposed method, it is not exploited for parameter efficiency. The reviewers were generally positive about the contribution, praised the clear exposition and promising results. However, there were concerns from reviewers xHEb and vB7z about the scalability of the method on larger LLMs (since the method has only been tested on GPT2 and ViT architectures in the original submission).  Reviewer xHEb was also concerned about the novelty of the approach. Whilst I agree that the idea is very simple and is a low-hanging fruit, the successful experimental execution even at the scale provided is a nontrivial contribution. In addition, the authors have added experiments on larger scale LLMs in their response to Reviewer vB7z, which I strongly encourage them to include in the camera ready version.